# Mirusviruses link herpesviruses to giant viruses

Morgan Gaïa[1,2,9], Lingjie Meng[3,9], Eric Pelletier[1,2], Patrick Forterre[4,5], Chiara Vanni[6], Antonio Fernandez-Guerra[7], Olivier Jaillon[1,2], Patrick Wincker[1,2], Hiroyuki Ogata[3], Mart Krupovic[8] & Tom O. Delmont[1,2✉]

DNA viruses have a major influence on the ecology and evolution of cellular organisms[1-4], but their overall diversity and evolutionary trajectories remain elusive[5]. Here we carried out a phylogeny-guided genome-resolved metagenomic survey of the sunlit oceans and discovered plankton-infecting relatives of herpesviruses that form a putative new phylum dubbed *Mirusviricota*. The virion morphogenesis module of this large monophyletic clade is typical of viruses from the realm *Duplodnaviria*[6], with multiple components strongly indicating a common ancestry with animal-infecting *Herpesvirales*. Yet, a substantial fraction of mirusvirus genes, including hallmark transcription machinery genes missing in herpesviruses, are closely related homologues of giant eukaryotic DNA viruses from another viral realm, *Varidnaviria*. These remarkable chimaeric attributes connecting *Mirusviricota* to herpesviruses and giant eukaryotic viruses are supported by more than 100 environmental mirusvirus genomes, including a near-complete contiguous genome of 432 kilobases. Moreover, mirusviruses are among the most abundant and active eukaryotic viruses characterized in the sunlit oceans, encoding a diverse array of functions used during the infection of microbial eukaryotes from pole to pole. The prevalence, functional activity, diversification and atypical chimaeric attributes of mirusviruses point to a lasting role of *Mirusviricota* in the ecology of marine ecosystems and in the evolution of eukaryotic DNA viruses.

Most double-stranded DNA viruses are classified into two major realms: *Duplodnaviria* and *Varidnaviria*. *Duplodnaviria* comprises tailed bacteriophages and related archaeal viruses of the class *Caudoviricetes* as well as eukaryotic viruses of the order *Herpesvirales*. *Varidnaviria* includes large and giant eukaryotic DNA viruses from the phylum *Nucleocytoviricota* as well as smaller viruses with tailless icosahedral capsids[6]. The two realms were established on the basis of the non-homologous sets of virion morphogenesis genes (virion module), including those encoding the structurally unrelated major capsid proteins (MCPs) with the 'double jelly-roll' and HK97 folds in *Varidnaviria* and *Duplodnaviria*, respectively[6]. Both realms are represented across all domains of life, with the respective ancestors thought to date back to the last universal cellular ancestor[7].

Within *Duplodnaviria*, bacterial and archaeal members of the *Caudoviricetes* exhibit a continuous range of genome sizes, from about 10 kilobases (kb) to >700 kb, whereas herpesviruses, restricted to animal hosts, are more uniform with genomes in the range of 100–300 kb. Herpesviruses probably evolved from bacteriophages, but the lack of related viruses outside the animal kingdom raises questions regarding their exact evolutionary trajectory[5]. Members of the *Varidnaviria* also exhibit a wide range of genome sizes, from about 10 kb to >2 Mb,

but there is a discontinuity in the complexity between large and giant viruses of the *Nucleocytoviricota* phylum and the rest of varidnaviruses with genomes <50 kb. It has been suggested that *Nucleocytoviricota* have evolved from a smaller varidnavirus ancestor[8-10], but the complexification entailing acquisition of multiple informational genes (informational module) remains to be fully understood.

Viruses within *Caudoviricetes* and *Nucleocytoviricota* are prevalent in the sunlit ocean where they play a critical role in regulating the community composition and blooming activity of plankton[11-17]. Here we carried out a genome-resolved metagenomic survey of planktonic DNA viruses guided by the phylogeny of a single hallmark gene. The survey covers nearly 300 billion metagenomic reads from surface-ocean samples of the *Tara* Oceans expeditions[18-20]. We characterized and manually curated hundreds of population genomes that expand the known diversity of *Nucleocytoviricota*. However, most notably, our survey led to the discovery of plankton-infecting relatives of herpesviruses that form a putative new phylum we dubbed *Mirusviricota*. The mirusviruses share complex functional traits and are widespread in the sunlit oceans where they actively infect eukaryotes, filling a critical gap in our ecological understanding of plankton. Despite a clear evolutionary relationship to herpesviruses,

[1]Génomique Métabolique, Genoscope, Institut François Jacob, CEA, CNRS, Univ. Evry, Université Paris-Saclay, Evry, France. [2]Research Federation for the Study of Global Ocean Systems Ecology and Evolution, FR2022/Tara GOSEE, Paris, France. [3]Bioinformatics Center, Institute for Chemical Research, Kyoto University, Uji, Japan. [4]Institut de Biologie Intégrative de la Cellule (I2BC), CNRS, Université Paris-Saclay, Gif sur Yvette, France. [5]Département de Microbiologie, Institut Pasteur, Paris, France. [6]MARUM Center for Marine Environmental Sciences, University of Bremen, Bremen, Germany. [7]Lundbeck Foundation GeoGenetics Centre, GLOBE Institute, University of Copenhagen, Copenhagen, Denmark. [8]Institut Pasteur, Université Paris Cité, CNRS UMR6047, Archaeal Virology Unit, Paris, France. [9]These authors contributed equally: Morgan Gaïa, Lingjie Meng. ✉e-mail: tdelmont@genoscope.cns.fr

mirusviruses encode even more genes that have closely related homologues in *Nucleocytoviricota*. These remarkable chimaeric attributes of *Mirusviricota* connect two distantly related virus realms, providing key insights into the evolution of eukaryotic DNA viruses.

## Genomics of marine eukaryotic viruses

DNA-dependent RNA polymerase subunits A (RNApolA) and B (RNApolB) are evolutionarily informative gene markers occurring in most of the known DNA viruses infecting marine microbial eukaryotes[9,21], which until now included only *Nucleocytoviricota*. Here we carried out a comprehensive search for RNApolB genes from the euphotic zone of polar, temperate and tropical oceans using large co-assemblies from 798 metagenomes (total of 280 billion reads that produced about 12 million contigs longer than 2,500 nucleotides)[19,20] derived from the *Tara* Oceans expeditions[18]. These metagenomes encompass eight plankton size fractions ranging from 0.8 μm to 2,000 μm (Supplementary Table 1), all enriched in microbial eukaryotes[22,23]. We identified RNApolB genes in these contigs using a broad-spectrum hidden Markov model (HMM) profile and subsequently built a database of more than 2,500 non-redundant environmental RNApolB protein sequences (similarity <90%; Supplementary Table 2). Phylogenetic signal for these sequences not only recapitulated the considerable diversity of marine *Nucleocytoviricota*[24] but also revealed previously undescribed deep-branching lineages clearly disconnected from the three domains of life and other known viruses (Extended Data Fig. 1). We reasoned that these new clades represent previously unknown lineages of double-stranded DNA viruses.

We carried out a phylogeny-guided genome-resolved metagenomic survey focusing on the RNApolB of *Nucleocytoviricota* and new clades to delineate their genomic context (Supplementary Table 3). We characterized and manually curated 581 non-redundant *Nucleocytoviricota* metagenome-assembled genomes (MAGs) up to 1.45 Mb in length (average of about 270 kb) and 117 non-redundant MAGs up to 438 kb in length (average of about 200 kb) for the new clades. We incorporated marine *Nucleocytoviricota* MAGs from previous metagenomic surveys[11,12] and reference genomes from culture and cell sorting to construct a comprehensive database enriched in large and giant marine eukaryotic double-stranded DNA viruses (thereafter called the Global Ocean Eukaryotic Viral (GOEV) database; Supplementary Table 4). The GOEV database contains about 0.6 million genes and provides contextual information to identify main ecological and evolutionary properties of MAGs containing the new RNApolB clades.

## Discovery of a third *Duplodnaviria* phylum

The newly assembled *Nucleocytoviricota* MAGs contain most of the hallmark genes of this viral phylum, corresponding to the virion and informational modules[4,5] (Supplementary Table 4). They expand the known diversity of the *Imitervirales*, *Pandoravirales*, *Pimascovirales* and *Algavirales* orders within the class *Megaviricetes*. In addition, one of the new RNApolB clades exposed a putative new *Nucleocytoviricota* class-level group we dubbed *Proculviricetes*, which is represented by six MAGs exclusively detected in the Arctic and Southern Oceans (Fig. 1). The 111 MAGs from the remaining new RNApolB clades also contain key genes evolutionarily related to the *Nucleocytoviricota* informational module, including RNApolA and RNApolB, family B DNA polymerase (DNApolB) and the transcription factor II-S (TFIIS). Single-gene phylogenies place these MAGs in one (DNApolB) or multiple clades (RNApolA and RNApolB), always in between the known *Nucleocytoviricota* orders (Extended Data Fig. 2). Signal for TFIIS was weaker owing to its shorter length. Robust phylogenomic inferences of the concatenated four informational gene markers indicate that they represent a monophyletic viral clade with several hallmark genes closely related to, yet distinct from, those in the known

*Nucleocytoviricota* classes (Fig. 1). We dubbed viruses in this clade the mirusviruses (*mirus* is a Latin word for surprising or strange).

The mirusvirus MAGs are organized into seven distinct subclades, M1 to M7 (from the most to least populated), with M1 and M7 being represented by 41 MAGs and a single MAG, respectively (Fig. 2a and Supplementary Table 4). Notably, however, they were devoid of identifiable homologues of the *Nucleocytoviricota* virion module, including the double jelly-roll MCP. Instead, annotation of mirusvirus gene clusters using sensitive sequence and structure similarity searches (Methods) identified a distant homologue of HK97-fold MCPs occurring in most of these MAGs (Fig. 1 and Extended Data Fig. 3). The presence of this MCP fold, shared only with *Caudoviricetes* and *Herpesvirales*, indicates that mirusviruses belong to the realm *Duplodnaviria*. Consistent with the identification of this MCP, further comparisons of HMM profiles and predicted three-dimensional (3D) structures uncovered key remaining components of the *Duplodnaviria* virion module, including the terminase (ATPase–nuclease, key component of the DNA packaging machine), portal protein, capsid maturation protease and triplex capsid proteins 1 and 2 (Fig. 1, Extended Data Fig. 4 and Supplementary Table 5). The presence of the genes encoding these proteins in mirusviruses establishes that they are bona fide large DNA viruses capable of forming viral particles similar to those of previously known viruses in the realm *Duplodnaviria*. Notably, phylogenetic inferences of the mirusvirus HK97-fold MCP recapitulated the seven subclades initially identified on the basis of DNApolB, RNApolA, RNApolB and TFIIS (Fig. 2b), indicative of a coevolution of the virion and informational modules.

The extensive sequence divergences and length disparities for proteins of the virion module between mirusviruses, herpesviruses and *Caudoviricetes* (Supplementary Table 5) prevented meaningful phylogenetic inferences for the newly expanded realm *Duplodnaviria*. Nevertheless, multiple components of this module provided critical insights clarifying the evolutionary trajectory of mirusviruses. First, the two triplex capsid proteins, which form a heterotrimeric complex and stabilize the capsid shell through interactions with adjacent MCP subunits[25], are conserved across herpesviruses but are missing in *Caudoviricetes*. Second, in herpesvirus MCPs, the HK97-fold domain, referred to as the floor domain and responsible for capsid shell formation, is embellished with a 'tower' domain that projects away from the surface of the assembled capsid[26]. The tower domain is an insertion within the A subdomain of the core HK97 fold[26,27]. In mirusviruses, the MCP protein also contains an insertion within the A subdomain, albeit of substantially smaller size (Fig. 1 and Extended Data Figs. 3 and 4). This tower domain has not been thus far described for any member of the *Caudoviricetes*, including the so-called jumbo phages (that is, phages with a very large genome[28]). Overall, the triplex capsid proteins and the MCP tower represent hallmark traits pointing to a closer evolutionary relationship between mirusviruses and herpesviruses compared to their bacterial and archaeal relatives.

Phylogenetic inferences of the DNApolB gene using the GOEV database and a wide range of eukaryotic and additional viral lineages[29] supported the evolutionary distance of mirusviruses relative to all other known clades of double-stranded DNA viruses (Extended Data Fig. 5). The monophyletic mirusvirus DNApolB was positioned as a sister clade to *Herpesviridae*, and the two clades of eukaryotic *Duplodnaviria* were most closely related to eukaryotic Zeta-type and Delta-type DNApolB sequences, together forming a strongly supported clade distinct from the DNApolB of other viruses. Taken together, the considerable genetic distances between the virion modules of mirusviruses, *Caudoviricetes* and *Herpesvirales*, the distinct 3D structures of the mirusvirus MCP (see predicted 3D structure comparisons in Extended Data Fig. 4) and the DNApolB phylogenetic inferences firmly position mirusviruses within the realm *Duplodnaviria*, but outside the two previously characterized phyla *Uroviricota* (*Caudoviricetes*) and *Peploviricota* (herpesviruses), in a separate phylum we dubbed *Mirusviricota*.

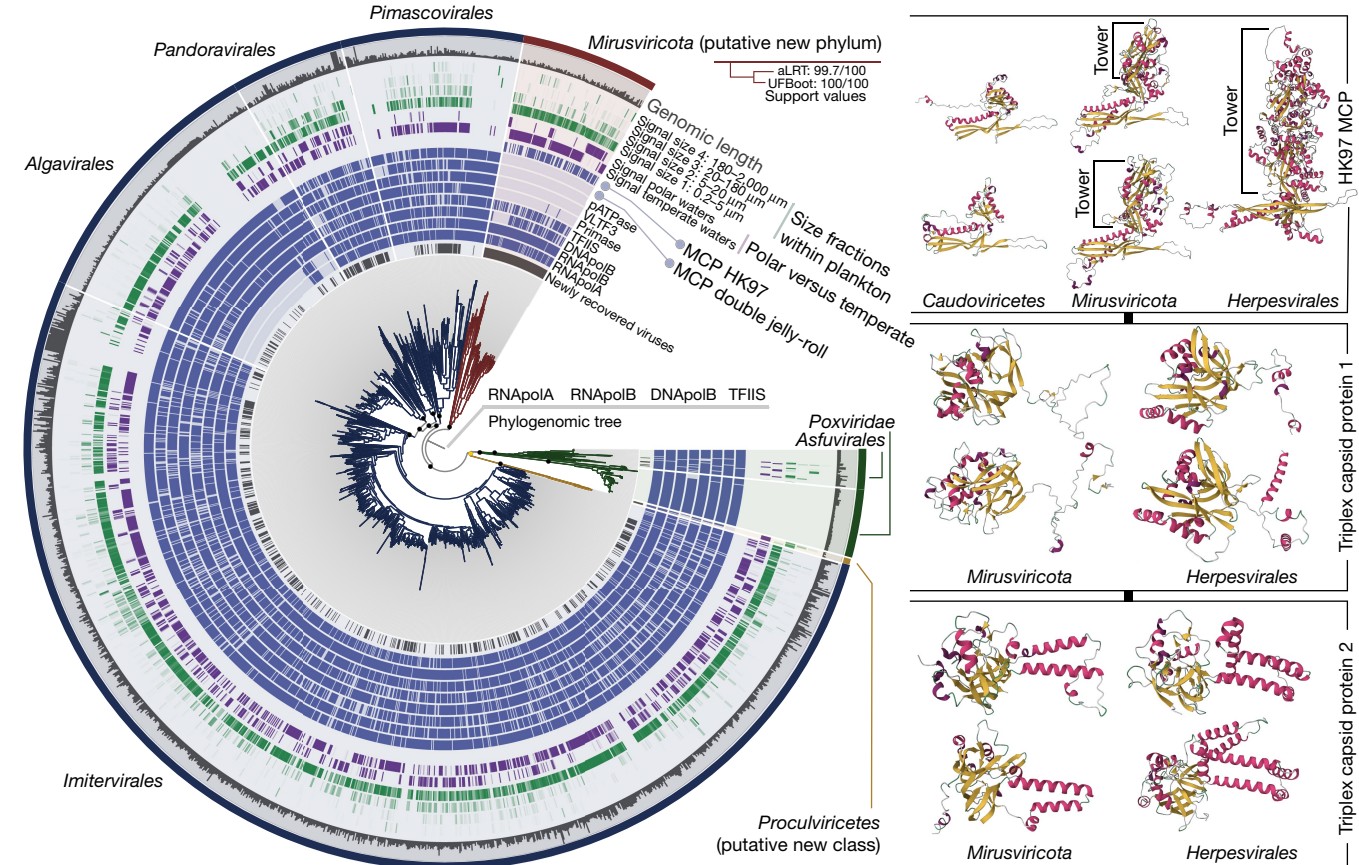

**Fig. 1 | Evolutionary relationships between *Nucleocytoviricota*, *Herpesvirales* and mirusviruses.** Left: a maximum-likelihood phylogenetic tree built from the GOEV database (1,722 genomes) on the basis of a concatenation of manually curated RNApolA, RNApolB, DNApolB and TFIIS genes (3,715 amino acid positions) using the posterior mean site frequency mixture model (LG + C30 + F + R10) and rooted between mirusviruses and the rest. Highlighted phylogenetic supports (dots in the tree) were considered high (approximation likelihood ratio (aLRT) ≥ 80 and ultrafast bootstrap approximation (UFBoot) ≥ 95, in black) or medium (aLRT ≥ 80 or UFBoot ≥ 95, in yellow; see Methods). The tree was decorated with rings of complementary information and visualized with anvi'o. Right: predicted 3D structures for the HK97 MCP of *Caudoviricetes*, mirusvirus and herpesvirus representatives obtained using AlphaFold2. Proteins are coloured on the basis of secondary structure properties. The panel also shows predicted 3D structures for the triplex capsid proteins of mirusvirus and herpesvirus representatives using the same methodology.

## Mirusviruses are functionally complex

The 111 *Mirusviricota* MAGs contain a total of 22,242 genes organized into 35 core gene clusters present in at least 50% of MAGs, 1,825 non-core gene clusters and finally 9,018 singletons with no close relatives within the GOEV database (Supplementary Tables 6 and 7). Core gene clusters provided a window into critical functional capabilities shared across subclades of mirusviruses (Supplementary Table 8). Aside from the aforementioned core components of the virion and informational modules, they correspond to functions related to DNA stability (H3 histone), DNA replication (DNA replication licensing factor, glutaredoxin/ribonucleotide reductase, Holliday junction resolvase and 3′ repair exonuclease 1), transcription (TATA-binding protein), gene expression regulation (lysine specific histone demethylase 1A), post-transcriptional modification of RNA (RtcB-like RNA-splicing ligase) and proteins (putative ubiquitin protein ligase), protein degradation (trypsin-like, C1 and M16-family peptidases), cell growth control (Ras-related protein), detection of external signals (sensor histidine kinase) and light-sensitive receptor proteins (heliorhodopsins). Thus, mirusviruses encode an elaborate toolkit that could enable fine-tuning the cell biology and energetic potential of their hosts for optimal virus replication. Finally, ten core gene clusters could not be assigned any function on the basis of sequence or structural comparisons to proteins in reference databases and await experimental functional characterization.

Clustering of *Mirusviricota* MAGs and reference viral genomes from culture (including *Nucleocytoviricota*, *Herpesvirales* and *Caudoviricetes*) based on quantitative occurrence of gene clusters highlighted the strong functional differentiation between mirusviruses and herpesviruses and, conversely, a strong functional similarity between mirusviruses and *Nucleocytoviricota* (Extended Data Fig. 6 and Supplementary Table 9). Thus, function-wise, mirusviruses more closely resemble the *Nucleocytoviricota* viruses (many of which are also widespread at the surface of the oceans; see Fig. 1) as compared to *Herpesvirales*. To further explore the functional landscape of eukaryote-infecting marine viruses, we clustered their genomes on the basis of quantitative occurrence of gene clusters using the entire GOEV database (Supplementary Tables 6 and 7). The mirusviruses clustered together and were further organized into subclades in line with phylogenomic signals (Extended Data Fig. 7). By contrast, this analysis emphasized the complex functional makeup of *Nucleocytoviricota* lineages, with some clades (for example, the *Imitervirales* and *Algavirales*) split into multiple groups. Aside from the core components of the informational module, gene clusters connecting a substantial portion of *Mirusviricota* and *Nucleocytoviricota* genomes were dominated by functions involved in DNA replication: the glutaredoxin/ribonucleotide reductase, Holliday junction resolvase, proliferating cell nuclear antigen, dUTPase and DNA topoisomerase II. Commonly shared functions also included the Ras protein, patatin-like phospholipase (lipid degradation), peptidase C1,

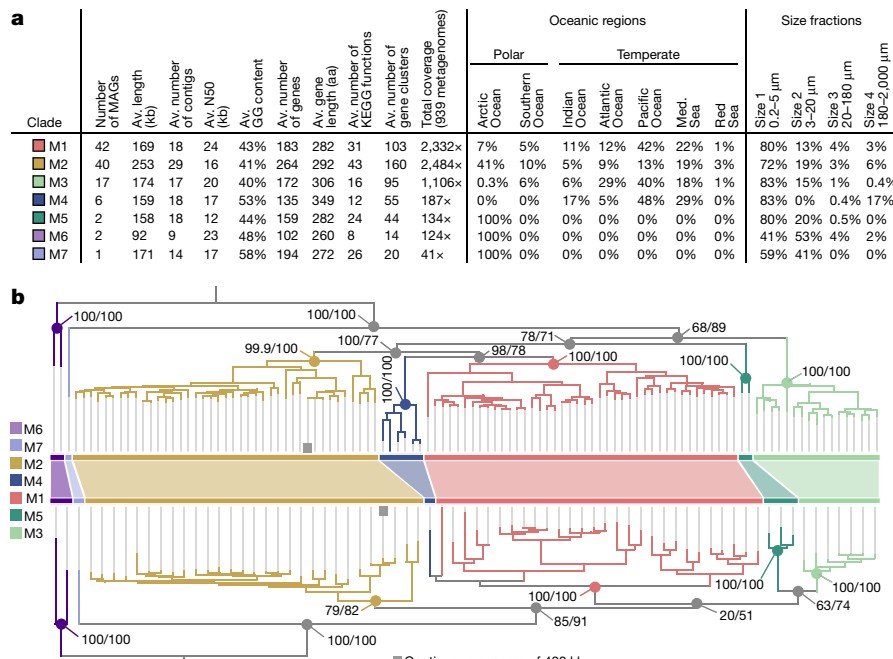

**a**

| Clade | Number of MAGs | Av. length (kb) | Av. number of contigs | Av. N50 (kb) | Av. GG content | Av. number of genes | Av. gene length (aa) | Av. number of KEGG functions | Av. number of gene clusters | Total coverage (939 metagenomes) | Oceanic regions | | | | | | | Size fractions | | | |
|---|---|---|---|---|---|---|---|---|---|---|---|---|---|---|---|---|---|---|---|---|---|
| | | | | | | | | | | | Polar | | Temperate | | | | | | | | |
| | | | | | | | | | | | Arctic Ocean | Southern Ocean | Indian Ocean | Atlantic Ocean | Pacific Ocean | Med. Sea | Red Sea | Size 1 0.2–5 µm | Size 2 3–20 µm | Size 3 20–180 µm | Size 4 180–2,000 µm |
| ■ M1 | 42 | 169 | 18 | 24 | 43% | 183 | 282 | 31 | 103 | 2,332× | 7% | 5% | 11% | 12% | 42% | 22% | 1% | 80% | 13% | 4% | 3% |
| ■ M2 | 40 | 253 | 29 | 16 | 41% | 264 | 292 | 43 | 160 | 2,484× | 41% | 10% | 5% | 9% | 13% | 19% | 3% | 72% | 19% | 3% | 6% |
| ■ M3 | 17 | 174 | 17 | 20 | 40% | 172 | 306 | 16 | 95 | 1,106× | 0.3% | 6% | 6% | 29% | 40% | 18% | 1% | 83% | 15% | 1% | 0.4% |
| ■ M4 | 6 | 159 | 18 | 17 | 53% | 135 | 349 | 12 | 55 | 187× | 0% | 0% | 17% | 5% | 48% | 29% | 0% | 83% | 0% | 0.4% | 17% |
| ■ M5 | 2 | 158 | 18 | 12 | 44% | 159 | 282 | 24 | 44 | 134× | 100% | 0% | 0% | 0% | 0% | 0% | 0% | 80% | 20% | 0.5% | 0% |
| ■ M6 | 2 | 92 | 9 | 23 | 48% | 102 | 260 | 8 | 14 | 124× | 100% | 0% | 0% | 0% | 0% | 0% | 0% | 41% | 53% | 4% | 2% |
| ■ M7 | 1 | 171 | 14 | 17 | 58% | 194 | 272 | 26 | 20 | 41× | 100% | 0% | 0% | 0% | 0% | 0% | 0% | 59% | 41% | 0% | 0% |

**Fig. 2 | Genomic statistics and evolution of mirusviruses. a**, Genomic and environmental statistics for the seven *Mirusviricota* subclades. Av., average; aa, amino acids; KEGG, Kyoto Encyclopedia of Genes and Genome; N50, the shortest contig length needed to capture 50% of the total assembly size; Med., Mediterranean. **b**, A maximum-likelihood phylogenetic tree built from the *Mirusviricota* MAGs on the basis of a concatenation of four hallmark informational genes (those encoding RNApolA, RNApolB, DNApolB and TFIIS; 3,715 amino acid positions) using the LG + F + R7 model. **c**, A maximum-likelihood phylogenetic tree built from the *Mirusviricota* MAGs on the basis of the MCP (701 amino acid positions) using the LG + R6 model. Both trees were rooted between clade M6 and other clades. Values at nodes represent branch supports (out of 100) calculated by the Shimodaira–Hasegawa-like aLRT (1,000 replicates; left score) and UFBoot (1,000 replicates; right score).

ubiquitin carboxy-terminal hydrolase (protein activity regulation) and the Evr1/Alr family (maturation of cytosolic Fe/S protein). Thus, the functional connectivity between the two phyla goes well beyond the informational module. On the other hand, hundreds of gene clusters and functions were significantly enriched in either mirusviruses or *Nucleocytoviricota* (Supplementary Tables 6–8), exposing distinct lifestyles for the two clades. Core gene clusters among the mirusviruses that were significantly less represented among *Nucleocytoviricota* genomes included the trypsin-like (73% of genomes in mirusviruses versus 9% in *Nucleocytoviricota*) and M16-family (60% versus 2%) peptidases, TATA-binding protein (59% versus 0%), heliorhodopsin (64% versus 5%) and histone (54% versus 2%). Phylogenetic inferences of the histones and rhodopsins point to a complex evolutionary history of these genes in both *Mirusviricota* and *Nucleocytoviricota*, with multiple horizontal transfer events between the virus clades and marine planktonic eukaryotes (Extended Data Fig. 8). In addition, a *Micromonas* heliorhodopsin may have originated from a mirusvirus (Extended Data Fig. 8), suggesting that *Mirusviricota* contributes, alongside *Nucleocytoviricota*[3,4], to the evolution of planktonic eukaryotes by means of gene flow.

## Mirusviruses are abundant and active

To our knowledge, *Mirusviricota* represents the first eukaryote-infecting lineage of *Duplodnaviria* found to be widespread and abundant within plankton in the sunlit oceans. Indeed, mirusviruses were detected in 131 out of the 143 *Tara* Oceans stations, from pole to pole. They occurred mostly in the 0.2–5 µm (76.3% of the entire mirusvirus metagenomic signal) and 3–20 µm (15.4%) size fractions that cover a high diversity of unicellular planktonic eukaryotes[22] (Figs. 1 and 2 and Supplementary Table 10). Among the *Tara* Oceans metagenomes considered in our study, the total mean coverage of marine *Nucleocytoviricota* MAGs and

culture genomes in GOEV was 15 times higher compared to that of the mirusvirus MAGs, reflecting the current imbalance in genomic units between these two phyla (1,706 versus 111). Yet, median cumulative mean coverage for the mirusviruses was higher compared to that for viruses in all *Nucleocytoviricota* orders, with the noticeable exception of *Algavirales* (Extended Data Fig. 9 and Supplementary Table 10). Thus, the mirusviruses are among the most abundant eukaryotic viruses characterized so far in the sunlit oceans.

The mirusviruses are not only abundant but also highly active within plankton. In fact, the mirusvirus MAGs, which contain just 3.8% of genes in GOEV, represent 13% of the *Tara* Oceans metatranscriptomic signal for this genomic database (Supplementary Table 11). This substantial in situ transcriptomic signal stresses the relevance of *Mirusviricota* to eukaryotic virus–host dynamics in marine systems. Mirusviruses were most active in the sunlit ocean (and especially in the euphotic subsurface layer enriched in chlorophyll) as compared to the mesopelagic zone (>200 m in depth), and within the cellular range of 0.2–20 µm (Fig. 3), in line with the metagenomic signal. The 35 core gene clusters for *Mirusviricota* represented 20% of the metatranscriptomic signal (including 12% for just seven capsid proteins), with remaining signal linked to non-core gene clusters (43%) and singletons (37%). Thus, highly diversified genes (nearly 10,000 singletons were identified) seem to play a critical role in the functional activity of *Mirusviricota* during infection of marine microbial eukaryotes.

Mirusviruses have different biogeographic distributions (for example, some are found only in the Arctic Ocean), yet their 35 core genes were expressed with similar levels in samples with metatranscriptomic signal, indicating a relatively homogeneous functional lifestyle regardless of latitude or subclade (Fig. 3 and Supplementary Table 11). The highest levels of expression were in genes coding for the capsid proteins, with ratios recapitulating the proportion of corresponding proteins in the capsid of herpesviruses (for example, more HK97 MCPs as compared

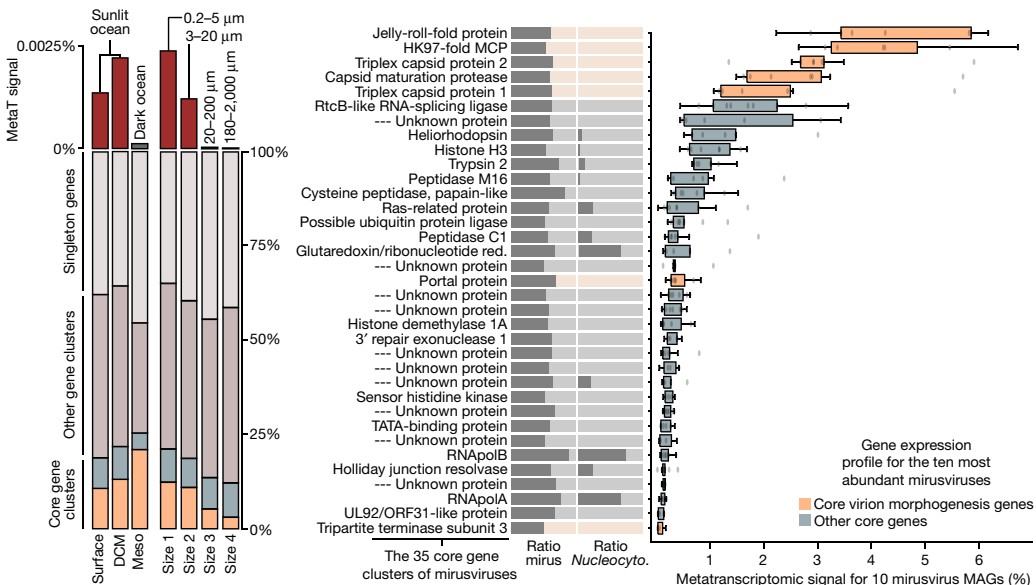

**Fig. 3 | In situ expression profile of mirusviruses during infection.**
Left: summary of the overall metatranscriptomic signal of different gene categories for the mirusvirus MAGs among the *Tara* Oceans metatranscriptomes. DCM, deep chlorophyll maximum layer; Meso, mesopelagic (top dark ocean layer below 200 m). Right, summary of the occurrence of 35 *Mirusviricota* core gene gene clusters as a ratio for the mirusvirus MAGs (mirus) and *Nucleocytoviricota* (*Nucleocyto.*). The panel also shows box plots corresponding to the overall metatranscriptomic signal for genes corresponding to the 35 core gene clusters and occurring in the 10 most abundant mirusviruses among the *Tara* Oceans metagenomes. Percentage values are genome-centric and correspond to the percentage of mean coverage (sum across all the metatranscriptomes) of one gene when considering the cumulated mean coverage of all genes (sum across all the metatranscriptomes) found in the corresponding genome. Centre lines in box plots show the medians; box limits indicate the 25th and 75th percentiles; whiskers extend 1.5 times the interquartile range from the 25th and 75th percentiles; outliers are represented by dots (*n* = 10 points). Red., reductase.

to triplex or portal proteins). Genes coding for the new types of heliorhodopsin and histone were also expressed at high levels, pointing to an important functional role during infection. Collectively, the biogeographic and in situ transcriptomic patterns of mirusviruses suggest that they actively infect abundant marine unicellular eukaryotes in both temperate and polar waters.

## Mirusviruses connect two viral realms

To further validate the genomic content of mirusviruses and to exclude the possibility of artificial chimaerism, we created an HMM for the newly identified *Mirusviricota* MCP and used it as bait to search for complete genomes in additional databases. First, we found only two *Mirusviricota* MCPs in a comprehensive viral genomic resource from the <0.2 μm size fraction of the surface oceans (Global Ocean Virome 2)[16], suggesting that most virions in this clade are larger than 0.2 μm in size. We subsequently screened for the *Mirusviricota* MCP in a database containing hundreds of metagenomic assemblies from the 0.2–3 μm size fraction of the surface oceans[30]. We found a contiguous *Mirusviricota* genome (355 genes) in the Mediterranean Sea affiliated to the clade M2 with a length of 431.5 kb, just 6 kb shorter than the longest *Mirusviricota* MAG (Fig. 2b,c). Its genes recapitulate the core functionalities of mirusviruses (for example, topoisomerase II, TATA-binding protein, histone, multiple heliorhodopsins, Ras-related GTPases, cell surface receptor, ubiquitin and trypsin), and 80 of these genes have a clear hit when compared to *Nucleocytoviricota* HMMs (see Methods and Extended Data Fig. 10). Most critically, not only are all hallmark genes for the informational (DNApolB, RNApolA, RNApolB and TFIIS) and virion (HK97-fold MCP, terminase, portal protein, capsid maturation protease and the two triplex capsid proteins) modules of *Mirusviricota* present but they also occur relatively homogeneously across the genome (Extended Data Fig. 10). Thus, this near-complete contiguous genome perfectly recapitulates the hallmark virion module traits shared only between

mirusviruses and herpesviruses, as well as the informational module shared between *Mirusviricota* and *Nucleocytoviricota* (Fig. 4).

On the one hand, mirusviruses belong to the realm *Duplodnaviria* on the basis of their virion module. On the other hand, their hallmark informational genes have homologues prevalent in the phylum *Nucleocytoviricota* with unexpectedly high levels of sequence similarity. These results strongly indicate that this informational module originated in either giant viruses (giant virus origin hypothesis; Fig. 4b) or mirusviruses (mirusvirus origin hypothesis; Fig. 4c) and was then transferred between their two realms, most likely after the long-lasting coevolution of the corresponding genes between viruses and proto-eukaryotic hosts[9]. Thus, the mirusviruses are not only integral components of the ecology of eukaryotic plankton, but they also fill critical gaps in our understanding of the evolutionary trajectories of two major realms of double-stranded DNA viruses.

## Discussion

Our phylogeny-guided genome-resolved metagenomic survey of plankton at the surface of five oceans and two seas exposed a major clade of large eukaryotic DNA viruses, with genomes that can reach more than 400 kb in length, which are diverse, prevalent and active in the sunlit oceans. This clade, dubbed *Mirusviricota*, corresponds to a putative new phylum within the realm *Duplodnaviria* that until now included only the bacteria- and archaea-infecting *Caudoviricetes* and animal-infecting *Herpesvirales*. The *Mirusviricota* phylum is organized into at least seven subclades that might correspond to distinct families. Although both mirusviruses and *Herpesvirales* are eukaryote-infecting duplodnaviruses, they exhibit very different genomic features. Most notably, mirusviruses substantially deviate from all other previously characterized groups of DNA viruses, with the virion morphogenesis module (the defining trait for highest-rank double-stranded DNA virus taxonomy) affiliated to the realm *Duplodnaviria* and the informational

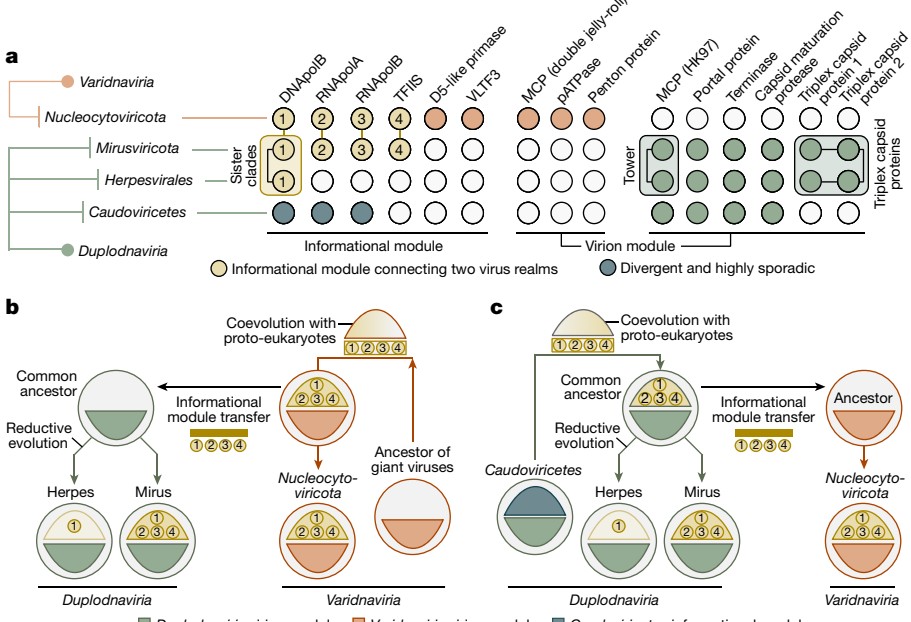

**Fig. 4 | Evolutionary trajectories of the eukaryotic informational module.**
**a**, Summary of the occurrence of hallmark genes for the informational and virion modules in *Nucleocytoviricota*, mirusviruses, herpesviruses and *Caudoviricetes*. Informational module genes with a strong evolutionary relationship are connected with a line. Genes containing information pointing to a common eukaryotic viral ancestry between mirusviruses and herpesviruses are framed. VLTF3, viral late transcription factor 3. **b**,**c**, Descriptions of two evolutionary scenarios in which the informational module of eukaryote-infecting viruses within the realms *Duplodnaviria* and *Varidnaviria* first emerged in the ancestor of either *Nucleocytoviricota* (giant virus hypothesis) or mirusviruses (mirusvirus hypothesis).

module closely related to that of large and giant viruses within the realm *Varidnaviria*. These apparent chimaeric attributes were recapitulated in a near-complete contiguous genome of 431.5 kb. The discovery of *Mirusviricota* is a reminder that we have not yet grasped the full ecological and evolutionary complexity of even the most abundant double-stranded DNA viruses in key ecosystems such as the surface of our oceans and seas.

Mirusviruses are relatively abundant in various regions of the sunlit oceans where they actively infect eukaryotic plankton smaller than 20 μm in size and express a variety of functions. *Mirusviricota* has a cohesive and complex inferred lifestyle that includes unique features (many core genes are found only in this phylum) but also substantially overlaps with those of large and giant eukaryotic varidnaviruses[11,12]. These shared functionalities go well beyond the informational module and include ecosystem- and host-specific genes, which could have been horizontally transferred between the two groups of viruses or convergently acquired from the shared hosts at different time points during evolution. For instance, the patatin-like phospholipase shared between the two phyla had already been suggested to promote the transport of *Nucleocytoviricota* genomes to the cytoplasm and nucleus[31]. Functions enriched in mirusviruses as compared to the *Nucleocytoviricota* include phylogenetically distinct H3 histones (proteins involved in chromatin formation within the eukaryotic cells[32]) and heliorhodopsins (light-sensitive receptor proteins that can be used as proton channels by giant viruses during infection[33]). Together, biogeographic patterns, functional gene repertoires and metatranscriptomic signal indicate that mirusviruses influence the ecology of key marine eukaryotes using a previously overlooked lifestyle.

Viruses of the *Herpesvirales* and *Nucleocytoviricota* belong to two ancient virus lineages, *Duplodnaviria* and *Varidnaviria*, respectively, with their corresponding ancestors possibly antedating the last universal cellular ancestor[6,7]. Nevertheless, the exact evolutionary trajectories and the identity of the respective most recent common ancestors of these prominent eukaryote-infecting double-stranded DNA viral

clades remain elusive, in part owing to the lack of known intermediate states. Particularly puzzling is the gap between the ubiquitous *Caudoviricetes*, some of which rival *Nucleocytoviricota* in terms of functional complexity and richness of their gene repertoires[34–36], and *Herpesvirales*, which are restricted to animal hosts and uniformly lack the transcription machinery and practice nuclear replication. The identification of *Mirusviricota* expands the presence of duplodnaviruses beyond animals to eukaryotic plankton hosts, strongly suggesting their ancient association with eukaryotes. The presence and location of the tower domain combined with the conservation of the two triplex capsid proteins (none of these is present in known *Caudoviricetes*) in both *Mirusviricota* and *Herpesvirales* (see Fig. 1) strongly suggests a common ancestry of these eukaryotic viruses, rather than independent evolution from distinct *Caudoviricetes* clades. The deep-branching positioning of mirusvirus informational genes attesting to one or multiple ancient transfers (Fig. 1 and Extended Data Fig. 2) and close similarity of the DNApolB between the two eukaryotic *Duplodnaviria* clades compared to other DNA virus clades (Extended Data Fig. 5) provide complementary information. With the shorter size of the tower domain and considering the later emergence of animals compared to unicellular eukaryotes, *Mirusviricota* viruses might more closely resemble the ancestral state of eukaryotic duplodnaviruses. Thus, mirusviruses point to a planktonic ancestry for herpesviruses, which would have undergone reductive evolution, most notably losing the transcription machinery, and specialized to the infection of animal cells[37].

Similarly enigmatic is the evolutionary trench between large and giant *Nucleocytoviricota* genomes and relatively simple varidnaviruses with modest gene repertoires for virion formation and genome replication (those infecting Bacteria and Archaea, as well as virophages, *Adenoviridae*, or else yaraviruses and polintoviruses[38,39]). It has been speculated that some of these simple varidnaviruses might represent evolutionary intermediates between bacteriophages and eukaryotic giant viruses from the phylum *Nucleocytoviricota*[5]. The genomic

complexity of mirusviruses within plankton, and their core functions shared with *Nucleocytoviricota* provide further insights. The informational module, and possibly other functions, may have been transferred from *Nucleocytoviricota* to the ancestor of mirusviruses (giant virus origin hypothesis), contributing to the complexification of eukaryotic duplodnaviruses. Under this scenario, a *Nucleocytoviricota* virus may have swapped its virion module with that of an uncharacterized duplodnavirus that co-infected the same host, while retaining the elaborate informational module. Yet, our data do not exclude the equally thought-provoking possibility of a transfer of the informational module from a mirusvirus to more simple ancestors of *Nucleocytoviricota* (mirusvirus origin hypothesis). This scenario could help explain the evolutionary leap from 'small' varidnaviruses to the overwhelmingly complex *Nucleocytoviricota*. Regardless of the hypothesis under consideration, mirusviruses clarify the evolutionary trajectory of eukaryotic double-stranded DNA viruses from both realms.

Overall, the prevalence, functional complexity and verified transcriptional activity of *Mirusviricota* point to a prominent role of the mirusviruses in the ecology of marine ecosystems. This putative phylum not only expands our understanding of plankton ecology, but it also provides new insights into virus evolution. Although the mirusviruses probably predated the emergence of herpesviruses, the timeline for *Mirusviricota* origins within plankton (before or after that of giant eukaryotic viruses) has yet to be elucidated. Moving forward, additional functional and genomic characterizations coupled with cultivation and environmental cell sorting for host identification will further contribute to our assessment of the lifestyle and prominence of mirusviruses within the oceans and beyond.

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

## Methods

### *Tara* Oceans metagenomes and metatranscriptomes

We analysed 937 metagenomes and 1,149 metatranscriptomes from *Tara* Oceans available at the EBI under project PRJEB402. Supplementary Tables 1 and 11 report general information (including the number of reads and environmental metadata) for each metagenome and metatranscriptome.

### Constrained automatic binning with CONCOCT

The 798 metagenomes corresponding to size fractions ranging from 0.8 μm to 2 mm were previously organized into 11 'metagenomic sets' on the basis of their geographic coordinates[19,20]. Those 0.28 trillion reads were used as inputs for 11 metagenomic co-assemblies using MEGAHIT[40] v1.1.1, and the contig header names were simplified in the resulting assembly outputs using anvi'o[41,42] v6.1. Co-assemblies yielded 78 million contigs longer than 1,000 nucleotides for a total volume of 150.7 Gb (refs. 19,20). Constrained automatic binning was carried out on each co-assembly output, focusing only on the 11.9 million contigs longer than 2,500 nucleotides. Briefly: anvi'o profiled contigs using Prodigal[43] v2.6.3 with default parameters to identify an initial set of genes; we mapped short reads from the metagenomic set to the contig using BWA v0.7.15 (ref. 44; minimum identity of 95%) and stored the recruited reads as BAM files using samtools[45]; anvi'o profiled each BAM file to estimate the coverage and detection statistics of each contig, and combined mapping profiles into a merged profile database for each metagenomic set. We then clustered contigs with the automatic binning algorithm CONCOCT[46] by constraining the number of clusters per metagenomic set to a number ranging from 50 to 400 depending on the set (total of 2,550 metagenomic blocks from about 12 million contigs)[19,20].

### Diversity of DNA-dependent RNApolB genes

We used HMMER[47] v3.1b2 to detect genes matching to the DNA-dependent RNApolB among all 2,550 metagenomic blocks on the basis of a single HMM model. We used CD-HIT[48] v4.8.1 to create a non-redundant database of RNApolB genes at the amino acid level with sequence similarity <90% (longest hit was selected for each cluster). Short sequences were excluded. Finally, we included reference RNApolB amino acid sequences from Bacteria, Archaea, Eukarya and giant viruses[9]: the sequences were aligned with MAFFT[49] v7.464 and the FFT-NS-i algorithm with default parameters and trimmed at >50% gaps with Goalign v0.3.5 (https://www.github.com/evolbioinfo/goalign). We carried out a phylogenetic reconstruction using the best-fitting model according to the Bayesian information criterion from the ModelFinder[50] Plus option with IQ-TREE[51] v1.6.2. We visualized and rooted the phylogeny using anvi'o. This tree allowed us to identify RNApolB corresponding to the known classes of *Nucleocytoviricota*, as well as new RNApolB clades.

### Phylogeny-guided genome-resolved metagenomics

Each metagenomic block containing at least one of the RNApolB genes of interest (see previous section) was manually binned using the anvi'o interactive interface to specifically search for *Nucleocytoviricota* and mirusvirus MAGs. First, we used HMMER[47] v3.1b2 to identify 8 hallmark genes (8 distinct HMM runs within anvi'o) as well as 149 additional orthologous groups often found in reference *Nucleocytoviricota* viruses[9] (a single HMM run within anvi'o). The interface considers the sequence composition, differential coverage, GC content and taxonomic signal of each contig, and displayed the eight hallmark genes as individual layers as well 149 additional orthologous groups often found in reference *Nucleocytoviricota* viruses[9] as a single extra layer for guidance. During binning, no restriction was applied in term of number of *Nucleocytoviricota* core gene markers present, as long as the signal suggested the occurrence of a putative MAG. Note that whereas some metagenomic blocks contained a limited number of MAGs, others contained dozens. Finally, we individually refined all of the *Nucleocytoviricota* and mirusvirus MAGs >50 kb in length as outlined in ref. 52, and renamed contigs they contained according to their MAG ID.

### Creation of the GOEV database

In addition to the *Nucleocytoviricota* and mirusvirus MAGs characterized in our study, we included marine *Nucleocytoviricota* MAGs characterized using automatic binning in ref. 11 ($n = 743$) and ref. 12 ($n = 444$), in part using *Tara* Oceans metagenomes. We also incorporated 235 reference *Nucleocytoviricota* genomes mostly characterized by means of cultivation but also cell sorting within plankton[53]. We determined the average nucleotide identity of each pair of *Nucleocytoviricota* or mirusvirus MAGs using the dnadiff tool from the MUMmer package[54] v4.0b2. MAGs were considered redundant when their average nucleotide identity was >98% (minimum alignment of >25% of the smaller MAG in each comparison). Manually curated MAGs were selected to represent a group of redundant MAGs. For groups lacking manually curated MAGs, the longest MAG was selected. This analysis provided a non-redundant genomic database of 1,593 marine MAGs plus 224 reference genomes, named the GOEV database. We created a single contigs database for the GOEV database using anvi'o. Prodigal[43] was used to identify genes.

### Curation of hallmark genes

The amino acid sequence datasets for RNApolA, RNApolB, DNApolB and TFIIS were manually curated through BLASTp alignments (BLAST[55] v2.10.1) and phylogenetic reconstructions, as previously described for eukaryotic hallmark genes[20]. Briefly, multiple sequences for a single hallmark gene within the same MAG were inspected on the basis of their position in a corresponding single-protein phylogenetic tree generated using the same protocol as described above (section entitled Diversity of DNA-dependent RNApolB genes). The genome's multiple sequences were then aligned with BLASTp to their closest reference sequence, and to each other. In case of important overlap with >95% identity (probably corresponding to a recent duplication event), only the longest sequence was conserved; in case of clear split, the sequences were fused and accordingly labelled for further inspection. Finally, RNApolA and RNApolB sequences shorter than 200 amino acids were also removed, as well as DNApolB sequences shorter than 100 amino acids, and TFIIS sequences shorter than 25 amino acids. This step created a set of curated hallmark genes.

### Alignments, trimming and single-protein phylogenetic analyses

For each of the four curated hallmark genes, the sequences were aligned with MAFFT[49] v7.464 and the FFT-NS-i algorithm with default parameters. Sites with more than 50% gaps were trimmed using Goalign v0.3.5 (https://www.github.com/evolbioinfo/goalign). The L-INS-i algorithm of MAFFT and a 70% threshold for trimming gappy sites were used for the MCP sequences of mirusviruses, the heliorhodopsin and the histone sequences (for the heliorhodopsin and histone, sequences from ref. 20 and additional histone reference sequences from ref. 56 were added). IQ-TREE[51] v1.6.2 was used for the phylogenetic reconstructions, with the ModelFinder[50] Plus option to determine the best-fitting model according to the Bayesian information criterion. Supports were computed from 1,000 replicates for the Shimodaira–Hasegawa (SH)-like aLRT[57] and UFBoot[58]. As per the IQ-TREE manual, supports were deemed good when SH-like aLRT ≥ 80% and UFBoot ≥ 95%. Anvi'o v7.1 was used to visualize and root the phylogenetic trees. The trees in Extended Data Fig. 2 do not include ambiguous genomes identified iteratively with the single and concatenated proteins phylogenies (see the section describing the supermatrix phylogenetic analysis). For the large DNApolB analysis, *Duplodnaviria* and *Baculoviridae* sequences from the National Center for Biotechnology Information (NCBI) viral genomic database (https://www.ncbi.nlm.nih.gov/labs/virus/vssi/#/; accessed

April 2022), as well as eukaryotic and viral sequences from ref. 29, were collected, aligned and trimmed, and the tree was reconstructed, with the same approaches as described, except for the FFT-NS-i algorithm used with MAFFT and the gap threshold set to 50% for Goalign. Very distant clades were iteratively removed, as well as long branches and phylogenetically uninformative sequences estimated with Treemmer[59], on the basis of a relative tree length of 0.95.

### Resolving hallmark genes occurring multiple times

We manually inspected all of the duplicated sequences (hallmark genes detected multiple times in the same genome) that remained after the curation step, in the context of the individual phylogenetic trees (see previous section). First, duplicates were treated as putative contaminations on the basis of major individual (that is, not conserved within a clade) incongruences with the position of the corresponding genome in the other single-protein trees. The putative contaminants were easily identified and removed. Second, we identified hallmark gene paralogues encapsulating entire clades and/or subclades, suggesting that the duplication event occurred before the diversification of the concerned viral clades. This is notably the case for most *Imitervirales*, which have two paralogues of the RNApolB. These paralogues were conserved for initial single-protein phylogenetic inferences, but then only the paralogue clades with the shortest branch were conserved for subsequent analyses, from single-protein trees to congruence inspection and concatenation. Finally, we also detected a small clade of *Algavirales* viruses containing a homologue of TFIIS branching distantly from the ordinary TFIIS type, suggesting a gene acquisition. These sequences were not included in subsequent analyses. This step created a set of curated and duplicate-free hallmark genes.

### Supermatrix phylogenetic analysis of the GOEV database

Concatenations of the four aligned and trimmed curated and duplicated-free hallmark genes (methods as described above) were carried out to increase the resolution of the phylogenetic tree. Genomes containing only TFIIS out of the four hallmark genes were excluded. For the remaining MAGs and reference genomes, missing sequences were replaced with gaps. Ambiguous genomes determined on the basis of the presence of major and isolated (that is, not a clade pattern) incongruences within single and concatenated protein trees, as well as on frequent long branches and unstable positions in taxon sampling inferences, were removed. The concatenated phylogenetic trees were reconstructed using IQ-TREE[51] v1.6.2 with the best-fitting model according to the Bayesian information criterion from the ModelFinder[50] Plus option. For the analysis including the entire GOEV database, the resulting tree was then used as a guide tree for a phylogenetic reconstruction based on the site-specific frequency posterior mean site frequency mixture model[60] (LG + C30 + F + R10). For the concatenated trees, supports were computed from 1,000 replicates for the SH-like aLRT[57] and UFBoot[58]. As per the IQ-TREE manual, supports were deemed good when SH-like aLRT ≥ 80% and UFBoot ≥ 95%. Anvi'o v7.1 was used to visualize and root the phylogenetic trees.

### Taxonomic inference of GOEV database

We determined the taxonomy of *Nucleocytoviricota* MAGs on the basis of the phylogenetic analysis results, using guidance from the reference genomes within the GOEV database as well as previous taxonomical inferences made in refs. 11,12,21.

### Biogeography of the GOEV database

We carried out a mapping of all metagenomes to calculate the mean coverage and detection of the GOEV database. Briefly, we used BWA v0.7.15 (minimum identity of 95%) and a FASTA file containing the 1,593 MAGs and 224 reference genomes to recruit short reads from all 937 metagenomes. We considered MAGs were detected in a given filter when >25% of their length was covered by reads to minimize non-specific read

recruitments[61]. The number of recruited reads below this cutoff was set to 0 before determining vertical coverage and percentage of recruited reads.

### Metatranscriptomics of the GOEV database

We carried out a mapping of all *Tara* Oceans metatranscriptomes to calculate the mean coverage and detection of genes found in the GOEV database. Briefly, we used BWA v0.7.15 (minimum identity of 95%) and a FASTA file containing the 0.6 million genes to recruit short reads from all 937 metagenomes.

### Orthologous groups from Orthofinder

Orthologous groups (OGs) in mirusvirus MAGs ($n = 111$), a mirusvirus near-complete contiguous genome and reference genomes from the Virus-Host Database (VHDB; including 1,754 *Duplodnaviria*, 184 *Varidnaviria* and 11 unclassified genomes) were generated. We used Orthofinder[62] v2.5.2 (-S diamond_ultra_sens) to generate OGs. A total of 26,045 OGs were generated and OGs ($n = 9,631$) with at least five genome observations were used to cluster genomes.

### AGNOSTOS functional aggregation inference

AGNOSTOS v.1 partitioned protein-coding genes from the GOEV database in groups connected by remote homologies and categorized those groups as members of the known or unknown coding sequence space on the basis of the workflow described previously[63]. AGNOSTOS produces groups of genes with low functional entropy as shown in refs. 20,63 allowing us to provide functional annotation (Pfam domain architectures) for some of the gene clusters using remote homology methods.

### Identification and modelling of the mirusvirus MCP

The putative MCP of mirusvirus and the other morphogenetic module proteins were identified with the guidance of AGNOSTOS results, using HHsearch against the publicly available Pfam v35, PDB70 and UniProt/Swiss-Prot viral protein databases[64,65]. The candidate MCP was then modelled using AlphaFold2 (refs. 66,67) (using Cobafold v1.4) and RoseTTAFold[68] v.1.1.0. The resulting 3D models were then compared to the MCP structures of phage HK97 and human cytomegalovirus and visualized using ChimeraX[69] v.1.4.

### Functional inferences of *Nucleocytoviricota* genomes

Genes from the GOEV database were BLASTp-searched against VHDB[70], RefSeq[71], UniRef90 (ref. 72), NCVOGs[73] (all databases were updated to the November 2021 version) and NCBI nr database (August 2020) using Diamond[74] v2.0.6 with a cutoff $E$ value $1 \times 10^{-5}$. A recently published GVOG database[21] was also used in annotation using hmmer[47] v3.2.1 search with an $E$ value of $1 \times 10^{-3}$ as a significant threshold. In addition, KEGG Orthology and functional categories were assigned with Eggnog-Mapper[75] v2.1.5. Finally, tRNAscan-SE[76] v2.0.7 predicted 7,734 tRNAs.

### 3D structure prediction of *Mirusviricota* core genes

Proteins corresponding to *Mirusviricota* core gene clusters and lacking functional annotation based on sequence similarities were modelled using AlphaFold2 v2.3.0 (refs. 66,67; -c full_dbs -t 2022-03-12). DALI server[77] was used to predict their functionality on the basis of protein structure comparisons.

### 3D structure prediction of *Duplodnaviria* hallmark virion module genes

Virion module genes of *Duplodnaviria* were collected from the NCBI protein database on the basis of the annotation in their initial submission. The genomes of virion module genes represent the viral families *Herpesviridae*, *Alloherpesviridae*, *Ackermannviridae*, *Autographiviridae*, *Chaseviridae*, *Demerecviridae*, *Drexlerviridae*, *Herelleviridae*, *Myoviridae*, *Podoviridae*, *Schitoviridae*, *Siphoviridae*, *Zobellviridae*, *Guelinviridae*, *Rountreeviridae*, *Salasmaviridae* and an unclassified

caudovirus, lilyvirus. The gene clusters of *Mirusviricota* corresponding to those virion modules were collected in seven mirusvirus subclades. The 3D models were predicted using AlphaFold2 v2.3.0 (refs. 66,67) (-c full_dbs -t 2022-03-12), and the first ranked structure model was used for the following analyses.

### 3D structure comparisons
Foldseek v4.645 (ref. 78) was used to align multiple predicted protein structures with the program easy-search. The TM score of the alignment was calculated and normalized by alignment length. The clustering of 3D structures for the *Duplodnaviria* MCP was carried out using the anvi'o programs anvi-matrix-to-newick and anvi-interactive with manual mode.

### Realm assignation of genes from a near-complete genome
Two in-house HMM databases were created as follows. First, all coding sequences (CDSs) labelled as *Nucleocytoviricota* were removed from the *Varidnaviria* CDS dataset (*n* = 53,776) in the VHDB[70] (May 2022). To this dataset, *Tara* Ocean *Nucleocytoviricota* MAGs (all were manually curated) and 235 reference *Nucleocytoviricota* genomes were integrated. The final *Nucleocytoviricota* protein database contained 269,523 CDSs. Similarly, we replaced all *Herpesvirales* CDSs in the VHDB *Duplodnaviria* CDS dataset with *Herpesvirales* protein sequences downloaded from NCBI in April 2022. Additionally, a marine *Caudovirales* database including jumbo phage environmental genomes[34,35] was integrated into the *Duplodnaviria* proteins. The final *Duplodnaviria* protein database contained 748,546 proteins. Proteins in the two databases were independently clustered at 30% sequence identity (-c 0.4 --cov-mode 5), using Linclust in MMseqs[79] v13-45111. Gene clusters with fewer than three genes were removed, and the remaining gene clusters were aligned using MAFFT[49] v7.487. HMM files (*n* = 16,689 and 57,259 for *Varidnaviria* and *Duplodnaviria*, respectively) were created using hmmbuild in HMMER3 (ref. 80) v3.2.1. All proteins in the near-complete *Mirusviricota* genome were searched against the two custom HMM databases using the hmmsearch with a cutoff *E* value of $1 \times 10^{-6}$.

### Statistical analyses
One-sided Fisher's exact test (greater) was used to identify KEGG Orthology functions as well as gene clusters with remote homologies that are significantly enriched in 111 *Mirusviricota* MAGs compared to all other *Nucleocytoviricota* in the GOEV database, on the basis of the occurrence of those functions and gene clusters. *P* values were corrected using the Benjamini–Hochberg procedure in R, and values <0.05 were considered significant.

### Naming of *mirus* and *procul*
The Latin adjective *mirus* (surprising, strange) was selected to describe the putative new *Duplodnaviria* phylum: the *Mirusviricota*. The Latin adverb *procul* (away, at distance, far off) was selected to describe the putative new class of *Nucleocytoviricota* discovered from the Arctic and Southern Oceans: the *Proculviricetes*.

### Reporting summary
Further information on research design is available in the Nature Portfolio Reporting Summary linked to this article.

## Data availability
Databases our study used include: *Tara* Oceans metagenomes and metatranscriptomes (https://www.ebi.ac.uk/ena/browser/view/PRJEB402); publicly available marine MAGs from the phylum *Nucleocytoviricota*[11,12]; the VHDB (https://www.genome.jp/virushostdb/); RefSeq (https://ftp.ncbi.nlm.nih.gov/refseq/); UniRef90 (https://ftp.ebi.ac.uk/pub/databases/uniprot/uniref/uniref90/); NCVOG (https://ftp.ncbi.nih.gov/pub/wolf/COGs/NCVOG/); and NCBI nr database (https://ftp.ncbi.nih.gov/blast/db/). Data generated in our study has been made publicly available at https://doi.org/10.6084/m9.figshare.20284713—this link provides access to: the RNApolB genes reconstructed from the *Tara* Oceans assemblies (along with references); individual FASTA files for the 1,593 non-redundant marine *Nucleocytoviricota* and mirusvirus MAGs (including the 697 manually curated MAGs from our survey) and 224 reference *Nucleocytoviricota* genomes contained in the GOEV database; the GOEV anvi'o contigs database; genes and proteins found in the GOEV database; manually curated hallmark genes; predicted 3D structures of the *Duplodnaviria* virion module (includes proteins and their alignments); phylogenies and associated anvi'o PROFILE databases with metadata; HMMs for hallmark genes; a FASTA file for the near-complete contiguous genome (SAMEA2619782_METAG_scaffold_2); and Supplementary Tables 1–11. Source data are provided with this paper.

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

**Acknowledgements** Our survey was made possible by two scientific endeavours: the sampling and sequencing efforts by the *Tara* Oceans Project, and the bioinformatics and visualization capabilities afforded by anvi'o (https://anvio.org/). We are indebted to all who contributed to these efforts, as well as other open-source bioinformatics tools for their commitment to transparency and openness. *Tara* Oceans (which includes the *Tara* Oceans and *Tara* Oceans Polar Circle expeditions) would not exist without the leadership of the *Tara* Oceans Foundation and the continuous support of 23 institutes (https://oceans.taraexpeditions.org/). We also acknowledge the commitment of the CNRS and Genoscope/CEA. Some of the computations were carried out using the platine, titane and curie high-performance computing machine provided through GENCI grants (t2011076389, t2012076389, t2013036389, t2014036389, t2015036389 and t2016036389). This study was supported in part by FRANCE GENOMIQUE (ANR-10-INBS-09), the Japan Society for the Promotion of Science KAKENHI (18H02279 and 22H00384), the Research Unit for Development of Global Sustainability, Kyoto University Research Coordination Alliance, and the International Collaborative Research Program of the Institute for Chemical Research, Kyoto University (2022-26, 2021-29 and 2020-28). M.K. was supported by grants from the l'Agence Nationale de la Recherche (ANR-20-CE20-0009-02 and ANR-21-CE11-0001-01), M.G. was supported by ANR ALGALVIRUS ANR-17-CE02-0012, and T.O.D. was supported by ANR HYDROGEN ANR-14-CE23-0001. Part of the computational work was carried out at the SuperComputer System, Institute for Chemical Research, Kyoto University. This article is contribution number 141 of *Tara* Oceans.

**Author contributions** T.O.D. conducted the study, which was initiated alongside M.G. and P.F. M.G., L.M., M.K., C.V., E.P. and T.O.D. carried out the primary data analysis. T.O.D. completed the genome-resolved metagenomic analysis. M.G. and T.O.D. curated the marker genes and identified the biological duplicates. M.G. carried out phylogenetic and phylogenomic analyses. L.M. carried out functional analyses, gene comparisons and protein structure predictions with the supervision of H.O. C.V. produced gene clusters with remote homologies with the supervision of A.F.-G. M.K. identified the MCP of *Mirusviricota* and other key genes of the virion module. E.P. carried out comparative genomic, biogeographic and metatranscriptomic analyses. All authors contributed to interpreting the data and writing the manuscript.

**Competing interests** The authors declare no competing interests.

**Additional information**
**Correspondence and requests for materials** should be addressed to Tom O. Delmont.

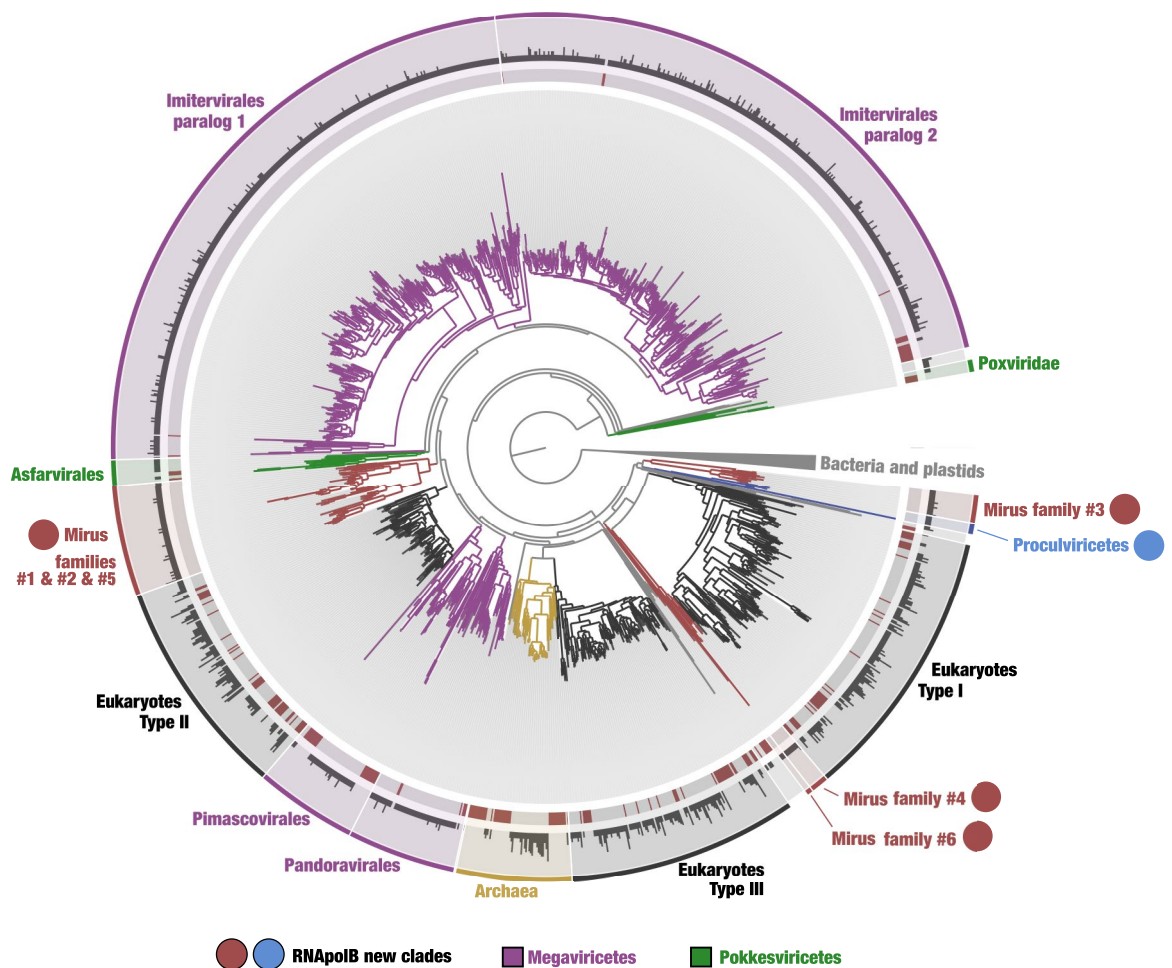

**Extended Data Fig. 1 | Identification of novel DNA-dependent RNA polymerase B (RNApolB) clades in the sunlit ocean.** The maximum-likelihood phylogenetic tree (LG+F+R10 model, 906 sites) is based on 2,728 RNApolB sequences more than 800 amino acids in length with similarity <90% (gray color in the inner ring) identified from 11 large marine metagenomic co-assemblies. This analysis also includes 262 reference RNApolB sequences (red color in the inner ring) corresponding to known archaeal, bacterial, eukaryotic and giant virus lineages for perspective. The middle ring shows the number of RNApolB sequences from the 11 metagenomic co-assemblies that match to the selected amino acid sequence with identity >90% (log10). The outer ring displays selections made for the different clades. Finally, RNApolB new lineages are labelled with a red dot for mirusviruses (subclades were characterized in subsequent analyses) and in blue for *Proculviricetes*.

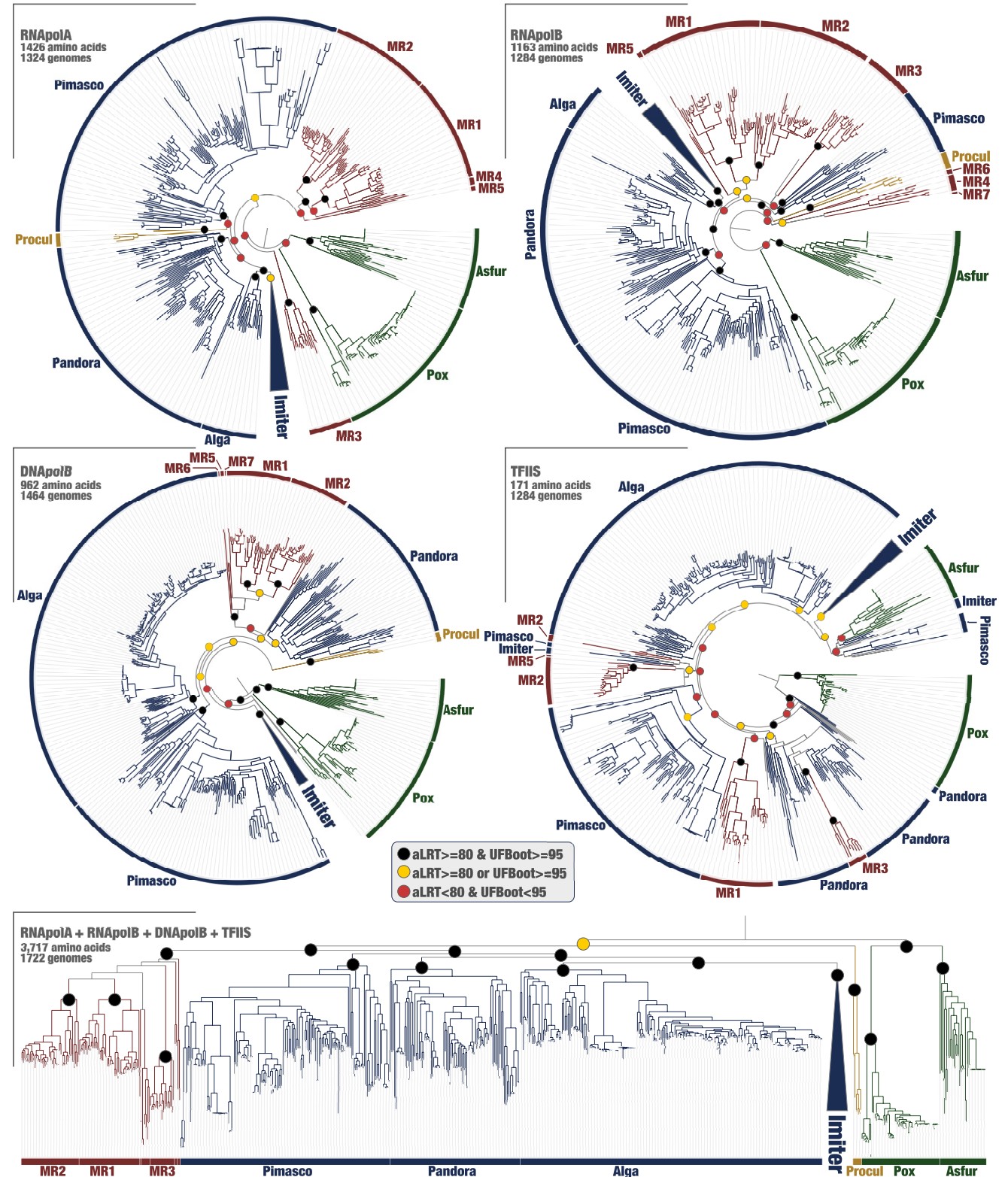

**Extended Data Fig. 2 | Single-protein and concatenated phylogenies of the four informational hallmark genes in the GOEV database.** Maximum-likelihood phylogenetic trees of the RNpolA, RNApolb, DNApolB and TFIIS were built from the GOEV database using the LG+F+R10 model (selected by ModelFinder Plus) and rooted between *Pokkesviricetes* and the rest.

Phylogenetic supports were considered high (aLRT>=80 and UFBoot>=95, in black), medium (aLRT>=80 or UFBoot>=95, in yellow) or low (aLRT<80 and UFBoot<95, in red) (see Methods). Finally, the concatenated tree described in Fig. 1 is also presented at the bottom for perspective.

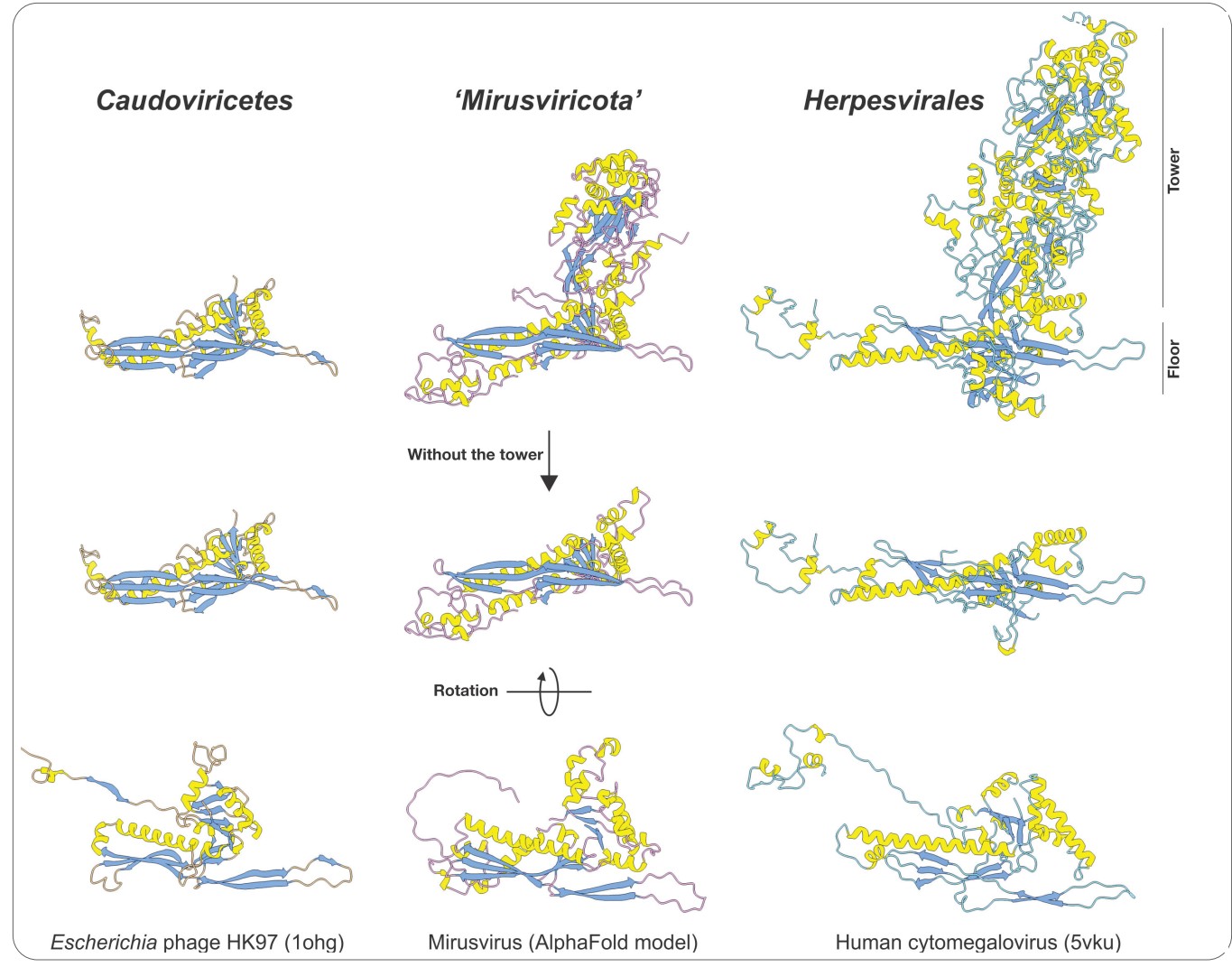

**Caudoviricetes**

**'Mirusviricota'**

**Herpesvirales**

Tower

Floor

Without the tower

Rotation

*Escherichia* phage HK97 (1ohg)

Mirusvirus (AlphaFold model)

Human cytomegalovirus (5vku)

**Extended Data Fig. 3 | 3D structure of the major capsid protein (MCP).**
The figure displays MCP 3D structures for *Escherichia* phage HK97
(*Caudoviricetes*), a representative genome for the mirusviruses (estimated using Alphafold), and the human cytomegalovirus (*Herpesvirales*). PDB accession numbers for the HK97 and cytomegalovirus MCPs are indicated in parentheses.

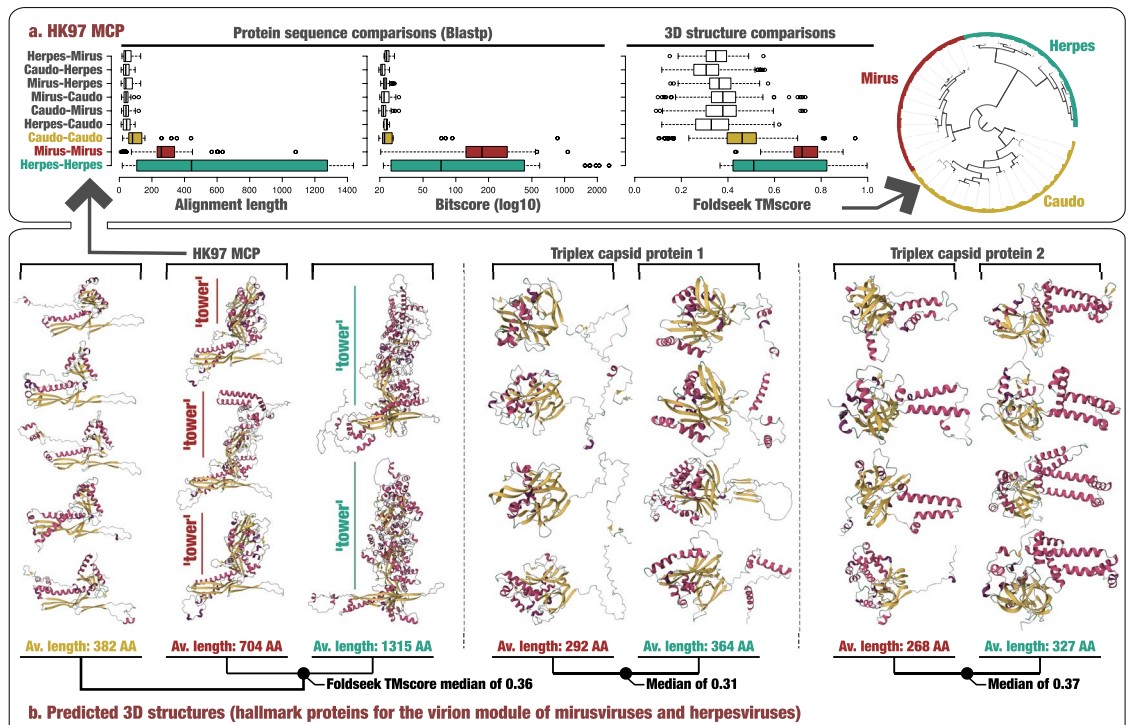

**Extended Data Fig. 4 | Protein sequence and predicated 3D structures comparisons.** Panel A displays protein sequence and 3D structure comparisons (Blastp and Foldseek) for the HK97 MCP of representatives covering various families from the three main *Duplodnaviria* clades. Center lines in boxplots show the medians; box limits indicate the 25th and 75th percentiles; whiskers extend 1.5 times the interquartile range from the 25th and 75th percentiles; outliers are represented by dots (from top to bottom, n = 22, 50, 38, 25, 35, 16, 40, 23 and 117 independent comparisons). The alignment values range from a minimum of 9 amino acids to a maximum of 1,437 amino acids. The bitscore values range from a minimum of 19.6 to a maximum of 2577. The Foldseek TMscore values range from a minimum of 0.09 to a maximum of 0.997. The dendrogram was generated using Euclidian distance and ward within anvi'o and is based on the Foldseek TMscore values. Panel B describes a selection of predicated 3D structures for the HK97 MCP and triplex proteins of representatives from the three main *Duplodnaviria* clades (*Caudoviricetes* viruses lack the triplex capsid proteins). Proteins are colored based on secondary structure properties.

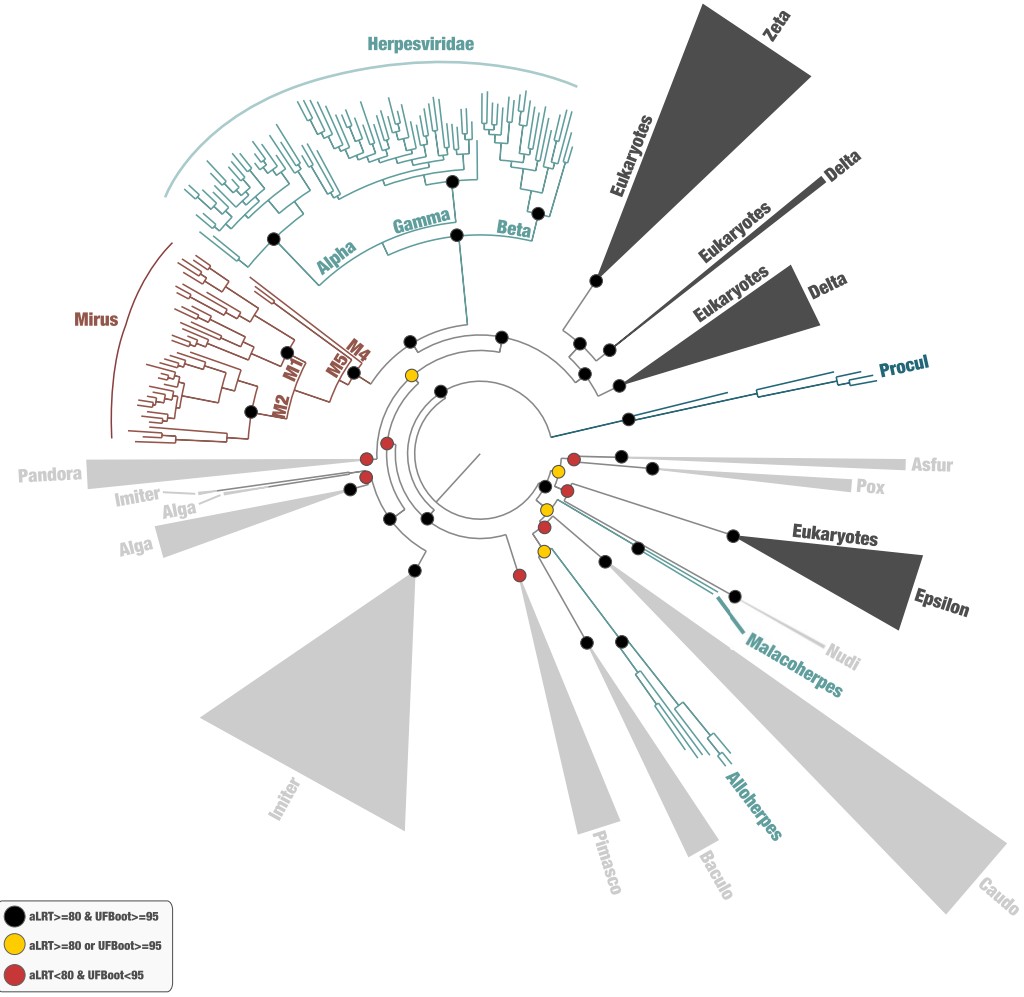

**Extended Data Fig. 5 | Phylogeny of the DNApolB hallmark gene.** The figure displays a maximum-likelihood phylogenetic tree (847 sites, 1,475 sequences) of DNA-polymerase B-family sequences using the LG+F+R10 model (selected by ModelFinder Plus) from the database described herein, *Duplodnaviria* and *Baculoviridae* sequences from the NCBI viral genomic database, and eukaryotic and viral sequences from Kazlauskas et al.[29] (see Methods). Eukaryotic Epsilon-type and related clades were used as outgroup. Phylogenetic supports were considered high (aLRT>=80 and UFBoot>=95, in black), medium (aLRT>=80 or UFBoot>=95, in yellow) or low (aLRT<80 and UFBoot<95, in red) (see Methods). Baculo: Baculoviridae; Caudo: Caudoviricetes; Nudi: Nudiviridae.

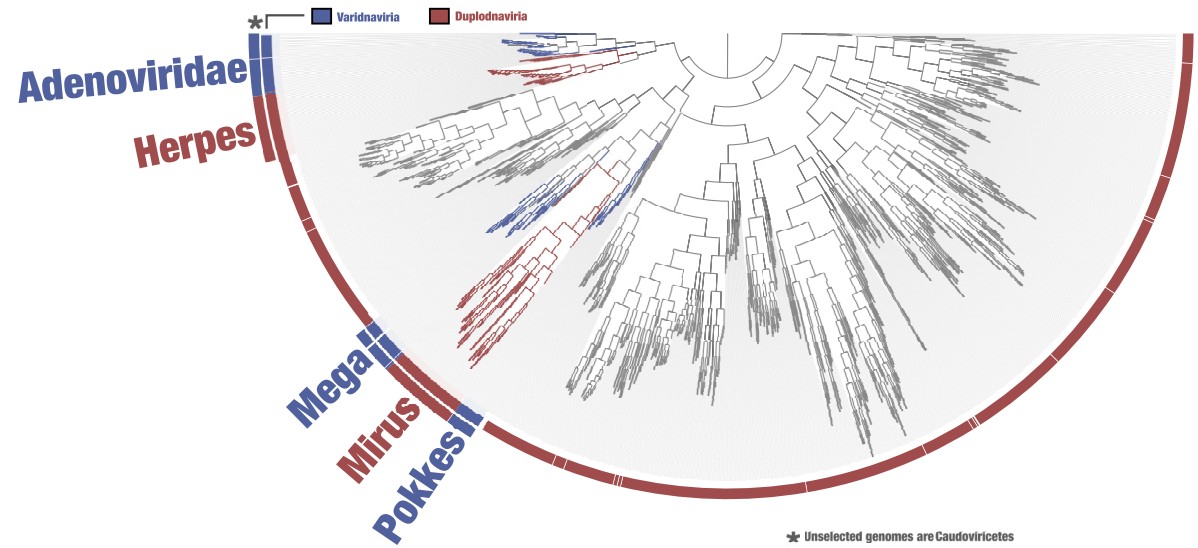

**Extended Data Fig. 6 | Functional clustering of mirusviruses and reference viral genomes from culture.** The inner tree is a clustering of '*Mirusviricota*' and other genomes based on the occurrence of all gene clusters (OrthoFinder method, Bray-Curtis distance).

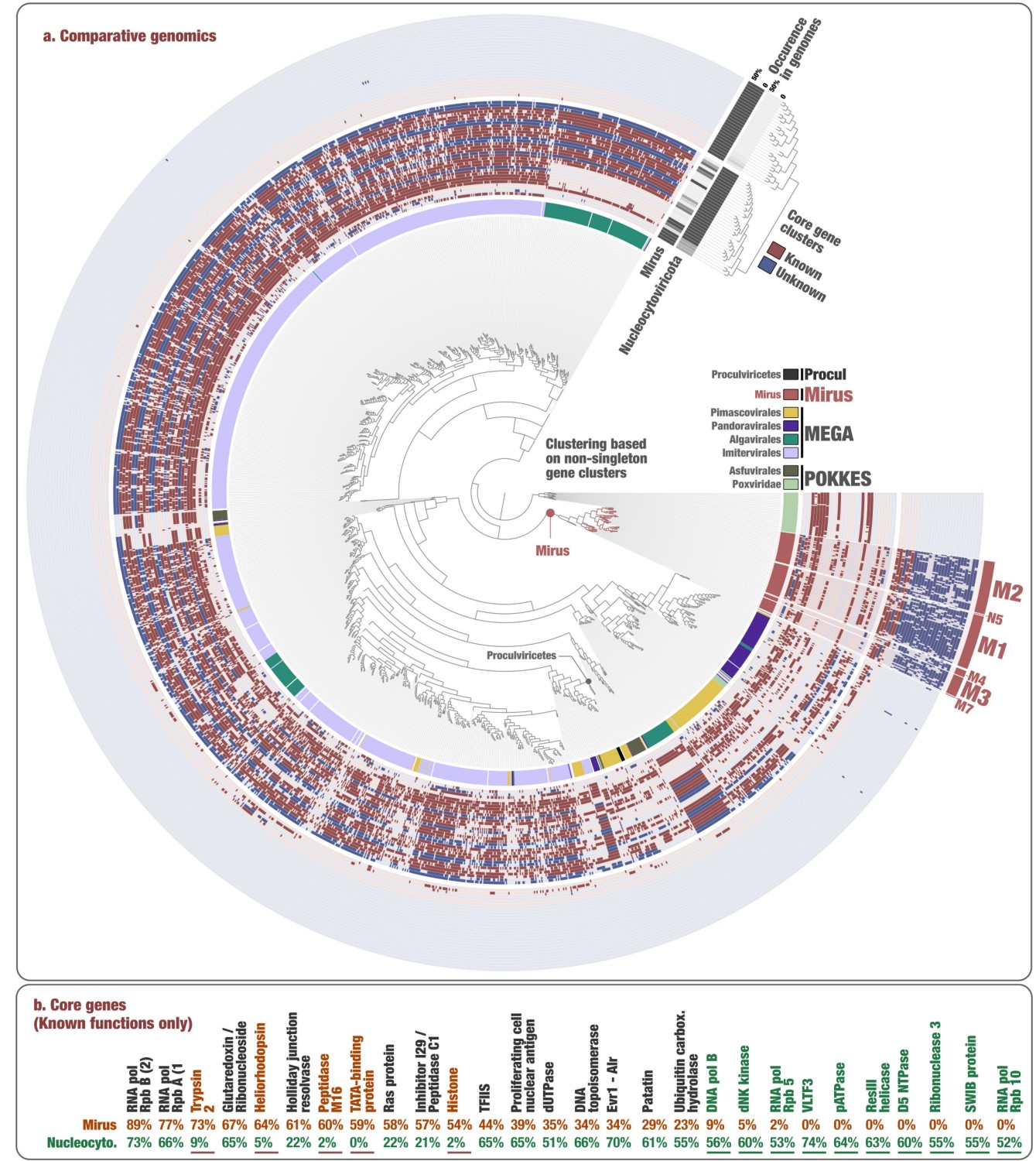

**a. Comparative genomics**

Occurrence in genomes

Core gene clusters
- Known
- Unknown

Clustering based on non-singleton gene clusters

Mirus

Proculviricetes

| Proculviricetes | Procul |
| Mirus | Mirus |
| Pimascovirales | |
| Pandoravirales | |
| Algavirales | MEGA |
| Imitervirales | |
| Asfuvirales | POKKES |
| Poxviridae | |

Mirus

Nucleocytoviricota

M2
N5
M1
M4
M3
M7

**b. Core genes (Known functions only)**

| | RNA pol Rpb B (2) | RNA pol Rpb A (1 | Trypsin 2 | Glutaredoxin / Ribonucleoside | Heliorhodopsin | Holliday junction resolvase | Peptidase M16 | TATA-binding protein | Ras protein | Inhibitor I29 / Peptidase C1 | Histone | TFIIS | Proliferating cell nuclear antigen | dUTPase | DNA topoisomerase | Evr1 - Alr | Patatin | Ubiquitin carbox. hydrolase | DNA pol B | dNK kinase | RNA pol Rpb 5 | VLTF3 | pATPase | ResIII helicase | D5 NTPase | Ribonuclease 3 | SWIB protein | RNA pol Rpb 10 |
|---|---|---|---|---|---|---|---|---|---|---|---|---|---|---|---|---|---|---|---|---|---|---|---|---|---|---|---|---|
| **Mirus** | 89% | 77% | 73% | 67% | 64% | 61% | 60% | 59% | 58% | 57% | 54% | 44% | 39% | 35% | 34% | 34% | 29% | 23% | 9% | 5% | 2% | 0% | 0% | 0% | 0% | 0% | 0% | 0% |
| **Nucleocyto.** | 73% | 66% | 9% | 65% | 5% | 22% | 2% | 0% | 22% | 21% | 2% | 65% | 65% | 51% | 66% | 70% | 61% | 55% | 56% | 60% | 53% | 74% | 64% | 63% | 60% | 55% | 55% | 52% |

**Extended Data Fig. 7 | Functional clustering of abundant and widespread marine viruses within mirusviruses and *Nucleocytoviricota*.** In panel A, the inner tree is a clustering of 'Mirusviricota' and *Nucleocytoviricota* genomes >100 kbp in length based on the occurrence of all the non-singleton gene clusters (Euclidean distance), rooted with the *Chordopoxvirinae* subfamily of *Poxviridae* genomes. Rings of information display the main taxonomy of *Nucleocytoviricota* as well as the occurrence of 60 gene clusters detected in at least 50% of 'Mirusviricota' or *Nucleocytoviricota*. The 60 gene clusters are clustered based on their occurrence (absence/presence) across the genomes. Panel B displays the occurrence of gene clusters of known Pfam functions detected in at least 50% of 'Mirusviricota' or *Nucleocytoviricota* genomes.

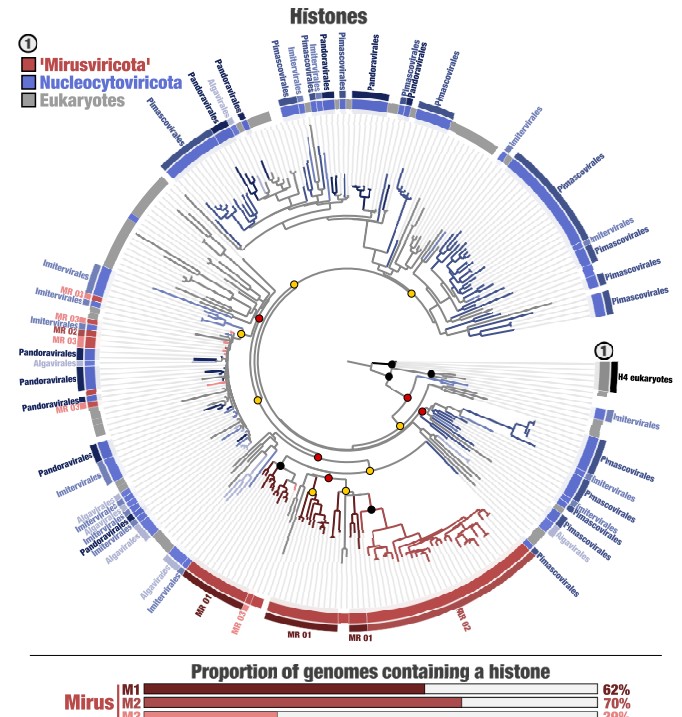

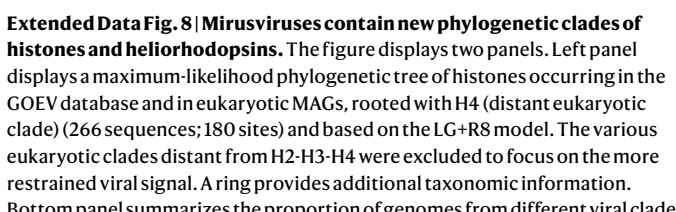

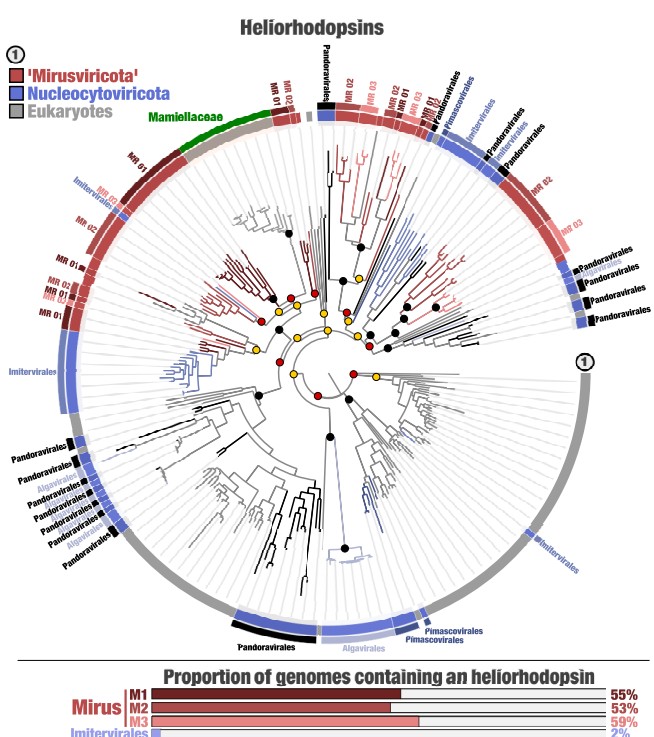

**Extended Data Fig. 8 | Mirusviruses contain new phylogenetic clades of histones and heliorhodopsins.** The figure displays two panels. Left panel displays a maximum-likelihood phylogenetic tree of histones occurring in the GOEV database and in eukaryotic MAGs, rooted with H4 (distant eukaryotic clade) (266 sequences; 180 sites) and based on the LG+R8 model. The various eukaryotic clades distant from H2-H3-H4 were excluded to focus on the more restrained viral signal. A ring provides additional taxonomic information. Bottom panel summarizes the proportion of genomes from different viral clades containing histones. Phylogenetic supports were considered high (aLRT>=80 and UFBoot>=95, in black), medium (aLRT>=80 or UFBoot>=95, in yellow) or

low (aLRT<80 and UFBoot<95, in red) (see Methods). Right panel displays a maximum-likelihood phylogenetic tree of heliorhodopsins occurring in the GOEV database and in eukaryotic MAGs (280 sequences; 313 sites), rooted with a large clade enriched in eukaryotes and based on the VT+F+R8 model. A ring provides additional taxonomic information. Bottom panel summarizes the proportion of genomes from different viral clades containing heliorhodopsins. Phylogenetic supports were considered high (aLRT>=80 and UFBoot>=95, in black), medium (aLRT>=80 or UFBoot>=95, in yellow) or low (aLRT<80 and UFBoot<95, in red) (see Methods).

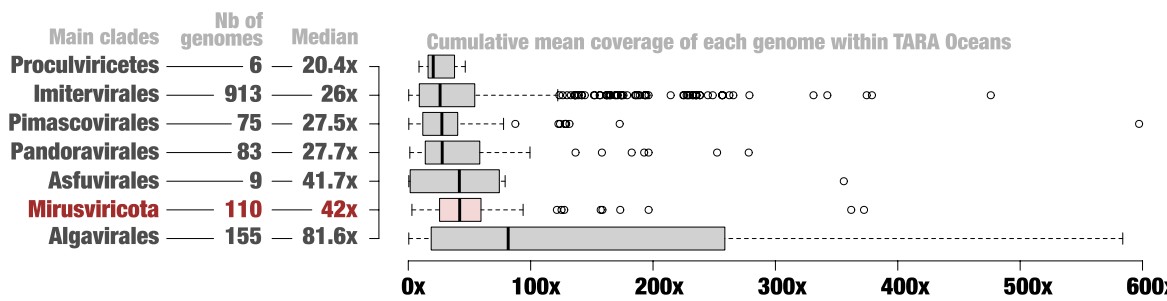

**Extended Data Fig. 9 | Environmental signal of virus eukaryotic clades in the sunlit oceans.** For each marine eukaryotic virus clades, the box plots display cumulative mean coverage of GOEV genomes among 937 *TARA* Oceans metagenomes. Only genome detected in at least one metagenome were considered. Center lines in boxplots show the medians; box limits indicate the 25th and 75th percentiles; whiskers extend 1.5 times the interquartile range from the 25th and 75th percentiles; outliers are represented by dots. The mean coverage values range from a minimum of 0.35x to a maximum of 6273.1x. The number of considered genomes per clade and their cumulative coverage median are also described.

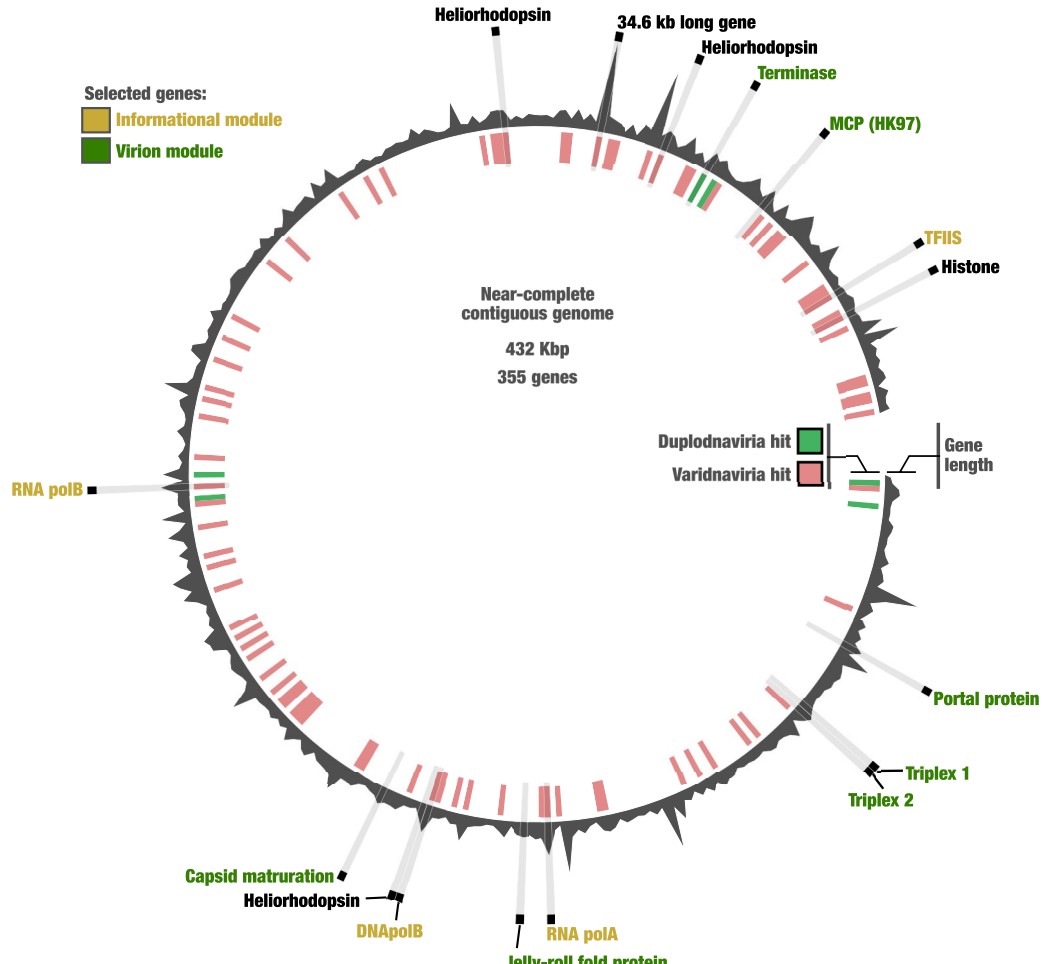

**Extended Data Fig. 10 | A near-complete genome for 'Mirusviricota'.** Synteny of 355 genes in the mirusvirus near-complete contiguous genome highlighting the occurrence of hallmark genes for the informational and virion modules, as well as heliorhodopsins and histone. Genes with a hit to HMMs from either *Duplodnaviria* or *Varidnaviria* are labelled in green and red, respectively (inner tree).

# Reporting Summary

## Statistics

For all statistical analyses, confirm that the following items are present in the figure legend, table legend, main text, or Methods section.

| n/a | Confirmed | |
|---|---|---|
| ☐ | ☒ | The exact sample size (*n*) for each experimental group/condition, given as a discrete number and unit of measurement |
| ☒ | ☐ | A statement on whether measurements were taken from distinct samples or whether the same sample was measured repeatedly |
| ☐ | ☒ | The statistical test(s) used AND whether they are one- or two-sided *Only common tests should be described solely by name; describe more complex techniques in the Methods section.* |
| ☒ | ☐ | A description of all covariates tested |
| ☒ | ☐ | A description of any assumptions or corrections, such as tests of normality and adjustment for multiple comparisons |
| ☒ | ☐ | A full description of the statistical parameters including central tendency (e.g. means) or other basic estimates (e.g. regression coefficient) AND variation (e.g. standard deviation) or associated estimates of uncertainty (e.g. confidence intervals) |
| ☐ | ☒ | For null hypothesis testing, the test statistic (e.g. *F*, *t*, *r*) with confidence intervals, effect sizes, degrees of freedom and *P* value noted *Give P values as exact values whenever suitable.* |
| ☒ | ☐ | For Bayesian analysis, information on the choice of priors and Markov chain Monte Carlo settings |
| ☒ | ☐ | For hierarchical and complex designs, identification of the appropriate level for tests and full reporting of outcomes |
| ☒ | ☐ | Estimates of effect sizes (e.g. Cohen's *d*, Pearson's *r*), indicating how they were calculated |

*Our web collection on statistics for biologists contains articles on many of the points above.*

## Software and code

Policy information about availability of computer code

| Data collection | no software was used to collect data |
|---|---|
| Data analysis | Genome-resolved metagenomics and visualizations were done using the platform anvi'o (v.7). HMMER (v3.1b2) was used to run Hidden Markov models. CD-HIT (v4.8.1) was used to remove protein redundancies. In order to perform phylogenetic analyses, we used MAFFT v7.464, Goalign v0.3.5, and IQ-TREE v1.6.2. Hallmark gene curations were performed using BLAST (v2.10.1). Metagenomic and metatranscriptomic read recruitments (mappping) were done using BWA v0.7.15. Both Orthofinder (v2.5.2), AGNOSTOS (v.1), and Linclust using MMseqs (v13-45111) were used to generate gene and protein clusters. Functional annotations were done using Pfam v35, PDB70, and UniProt/Swiss-Prot viral protein databases. They were also done using Virus-Host DB, RefSeq, UniRef90, NCVOGs (updated to the November 2021 version), and the NCBI nr database (August 2020), with using Diamond (v2.0.6). In addition, KEGG Orthology and functional categories were assigned with the Eggnog-Mapper (v2.1.5), and tRNAscan-SE75 (v2.0.7) was used to predict tRNAs. 3D structures were modeled using AlphaFold2 (v2.3.0) and RoseTTAFold v1.4. Foldseek (v4.64577) was used to compare protein 3D structures. |

For manuscripts utilizing custom algorithms or software that are central to the research but not yet described in published literature, software must be made available to editors and reviewers. We strongly encourage code deposition in a community repository (e.g. GitHub). See the Nature Portfolio guidelines for submitting code & software for further information.

## Data

Policy information about availability of data

All manuscripts must include a data availability statement. This statement should provide the following information, where applicable:

- Accession codes, unique identifiers, or web links for publicly available datasets
- A description of any restrictions on data availability
- For clinical datasets or third party data, please ensure that the statement adheres to our policy

Databases our study used include (1) the TARA Oceans metagenomes and metatranscriptomes (https://www.ebi.ac.uk/ena/browser/view/PRJEB402), (2) publicly available Nucleocytoviricota MAGs (https://www.nature.com/articles/s41586-020-1957-x and https://www.nature.com/articles/s41467-020-15507-2), (3) and Virus-Host DB (https://www.genome.jp/virushostdb/), (4) RefSeq (https://ftp.ncbi.nlm.nih.gov/refseq/), (5) UniRef90 (https://ftp.ebi.ac.uk/pub/databases/uniprot/uniref/uniref90/), (6) NCVOG (https://ftp.ncbi.nih.gov/pub/wolf/COGs/NCVOG/) and (7) NCBI nr database (https://ftp.ncbi.nih.gov/blast/db/). Data our study generated has been made publicly available at https://doi.org/10.6084/m9.figshare.20284713. This link provides access to (1) the RNApolB genes reconstructed from the Tara Oceans assemblies (along with references), (2) individual FASTA files for the 1,593 non-redundant marine Nucleocytoviricota and mirusvirus MAGs (including the 697 manually curated MAGs from our survey) and 224 reference Nucleocytoviricota genomes contained in the GOEV database, (3) the GOEV anvi'o CONTIGS database, (4) genes and proteins found in the GOEV database, (5) manually curated hallmark genes, (6) predicted 3D structures of the Duplodnaviria virion module (includes proteins and their alignments), (7) phylogenies and associated anvi'o PROFILE databases with metadata, (8) HMMs for hallmark genes, (9) a FASTA file for the near-complete contiguous genome (SAMEA2619782_METAG_scaffold_2), (10) and all the supplemental tables.

## Human research participants

Policy information about studies involving human research participants and Sex and Gender in Research.

| | |
|---|---|
| Reporting on sex and gender | Not applicable |
| Population characteristics | Not applicable |
| Recruitment | Not applicable |
| Ethics oversight | Not applicable |

Note that full information on the approval of the study protocol must also be provided in the manuscript.

# Field-specific reporting

Please select the one below that is the best fit for your research. If you are not sure, read the appropriate sections before making your selection.

☐ Life sciences  ☐ Behavioural & social sciences  ☒ Ecological, evolutionary & environmental sciences

For a reference copy of the document with all sections, see nature.com/documents/nr-reporting-summary-flat.pdf

# Ecological, evolutionary & environmental sciences study design

All studies must disclose on these points even when the disclosure is negative.

| | |
|---|---|
| Study description | The study is based on metagenomic data generated by the Tara Oceans consortium over the years. We characterized and manually curated environmental genomes for giant viruses as well as a previously unknown clade dubbed "Mirusviricota". |
| Research sample | Sunlit oceans (plankton) |
| Sampling strategy | The study did not involve any sampling, and we used data generated by the Tara Oceans consortium |
| Data collection | We used all Tara Oceans metagenomes. Those were generated in our institute s part of previous publications, so we had direct access to the data. The data is also publicly available to others. |
| Timing and spatial scale | The study did not involve any sampling or other data collection. |
| Data exclusions | No data was excluded. |
| Reproducibility | All data is available, and the tool anv'io is available for all to reproduce our findings. |
| Randomization | We worked on all the metagenomic legacy of Tara Oceans, so their is no randomization. |
| Blinding | We worked on all the metagenomic legacy of Tara Oceans, so their is no blinding. |

Did the study involve field work? ☐ Yes ☒ No

# Reporting for specific materials, systems and methods

We require information from authors about some types of materials, experimental systems and methods used in many studies. Here, indicate whether each material, system or method listed is relevant to your study. If you are not sure if a list item applies to your research, read the appropriate section before selecting a response.

## Materials & experimental systems

| n/a | Involved in the study |
|-----|------------------------|
| ☒ ☐ | Antibodies |
| ☒ ☐ | Eukaryotic cell lines |
| ☒ ☐ | Palaeontology and archaeology |
| ☒ ☐ | Animals and other organisms |
| ☒ ☐ | Clinical data |
| ☒ ☐ | Dual use research of concern |

## Methods

| n/a | Involved in the study |
|-----|------------------------|
| ☒ ☐ | ChIP-seq |
| ☒ ☐ | Flow cytometry |
| ☒ ☐ | MRI-based neuroimaging |

