## [Peer Review File · Nature]

Manuscript Title: Mirusviruses link herpesviruses to giant viruses

Reviewer Comments & Author Rebuttals

Reviewer Reports on the Initial Version:

Referee #2 (Remarks to the Author):

Gaia et al present a metagenomic study of large DNA viruses in the ocean. The authors use a binning approach to recover genomes of large DNA viruses, including NCLDVs, and discover a novel lineage that appears to be a chimera of NCLDV and herpesviruses. The morphogenetic module of these Mirusviruses (capsid, packaging ATPase) bears homology to Herpesviruses and tailed phages, while the rest of the genome (including the information module) is more similar to NCLDVs. Detailed bioinformatic and predicted structural analyses are performed to explore this novel lineage.

Overall this is an excellent study that uses creative bioinformatic methods to describe a remarkable new viral lineage that has so far eluded detection despite its widespread occurrence in the ocean. The discovery of a novel viral lineage with chimeric genomes is an important milestone, and the evolutionary analysis of mirusviruses and structural comparisons between capsids are by far the most significant components of this study. To my knowledge this is the first example of a truly chimeric dsDNA viral lineage that is the product of two distinct deep-branching lineages, and I agree with the authors that this represents a new phylum. This discovery is notable and will be of interest to a broad range of scientists, and as such warrants publication in a premier journal.

That said, I do have some general comments and constructive disagreement regarding the evolutionary interpretation that I hope the authors will consider. Below I have laid out my understanding of the facts and a possible alternative explanation for the cryptic evolutionary origins of the Mirusviruses.

-The most dramatic claim made in this study is that Mirusviruses are an ancient progenitor to either Herpesviruses or NCLDV, depending on the evolutionary scenario (Fig 5). The evidence that mirusviruses are a primordial progenitor to either NCLDV or herpesviruses is speculative, however, and other scenarios appear more likely to me. It is not more parsimonious to presume that mirusviruses are simply NCLDV that swapped out the usual structural components for those of a different viral realm? There are several reasons why this more recent emergence of mirusviruses appears more likely (to me at least): 1) Other NCLDVs have also acquired different capsids (pithoviruses, pandoraviruses) so this is actually rather common- the only difference here is that they acquired a HK97 capsid and associated genes. 2) Phylogenies of genes of the information module show that mirusviruses usually branch from within NCLDV (Fig S2), indicating that NCLDV already existed and had diversified by the time that Mirusviruses emerged. The PoIB tree with eukaryotic homologs (Fig S4) provides some tentative evidence of an independent Mirus origin, but this is not definitive because NCLDV have acquired this gene from eukaryotes multiple times themselves. 3) The phylogenetic breadth of the mirusviruses is not as broad as NCLDV, suggesting that they represent an order or class level of NCLDV that swapped out the morphogenetic module at

some point. 4) The other similarities of Mirusviruses and NCLDV, such as the common presence of histones, rhodopsins, and other functional genes also points to the overall NCLDV origin of mirusviruses. Indeed, the gene sharing tree also shows Mirusviruses emerging from within NCLDV 5) The rhodopsins and histones of Mirusviruses don't appear to be particularly basal branching compared to Mirusviruses- it's a bit hard to see because the NCLDV branches are not colored, but I don't see evidence of Mirusvirus ancestry here.

Now, the scenario in which mirusviruses are a primordial progenitor to herpesviruses is a bit more plausible, though I still do not favor this hypothesis. The main evidence for this is that the mirusvirus capsid is highly divergent from herpesvirus capsids, but still contains the tower domain. But this pattern could arise through many different processes - not necessarily only because mirusviruses gave rise to herpesviruses. For example, we don't know the evolutionary rates / mutation rate of mirusviruses - it may simply be higher than herpesviruses and NCLDV, which could explain a more rapid divergence. If this was the case, mirusviruses may have emerged from within herpesviruses (coinfection of a eukaryotic host may have been the initial site for recombination). Similarly, if the floor domain evolved convergently, then mirusviruses may have evolved from tailed phages, without any real link to herpesviruses. Based on the present data it is not clear if any of these alternative scenarios can be confidently ruled out.

-In my view, it would be helpful if more space and discussion were given to the significance of the Mirus capsid structure and other morphogenetic components (terminase, etc). This is undoubtedly the most important part of this paper, and the strongest evidence of genome chimerism, so a more detailed explanation of the floor vs tower domains could be given, together with their evolutionary significance. Is it possible that the tower domain could evolve convergently due to shared function in eukaryotic viruses? Does the structure necessarily imply shared evolutionary history? What else is known about the role of the tower domain? Right now the link between Mirusviruses and herpesviruses is somewhat tentative given the inability to produce molecular phylogenies. Predicted structures of the herpesvirus, phage, and mirusvirus packaging ATPase would also be welcome in the main text, given this is a major crux of the paper.

-Perhaps I missed it, but it would be useful if the authors could define the level of divergence between the herpesvirus, phage, and mirusvirus morphogenetic components a bit more. Is there no detectable homology of the mirusvirus proteins using BLASTP, Psiblast, or HMMER3? Perhaps an alignment could be made and put in the supplementary just to show that aligned blocks do not exist? The lack of detectable sequence homology between the mirusvirus proteins and other homologs is a key point of this paper, so adding something else in addition to statements in the text would be helpful.

-66-71: the whole premise of the ambiguity of how NCLDV emerged from smaller viruses is somewhat overstated, in my opinion. Many related smaller viruses have Double jelly roll fold capsids - namely polintoviruses, virophages, and most recently Yaravirus (10.1073/pnas.2001637117)- and they already have been proposed to be a link between NCLDV and smaller viruses (10.1016/j.virol.2015.02.039). The Woo et al study that is cited on line 70 nicely shows this too. Even if this premise were correct, it is not clear how mirusviruses necessarily solve this riddle given that their genomes are rather large (the complete genome is 435 kbp) - they are not really an

intermediate between NCLDV and smaller viruses. Once again, I feel that the logic pushing mirusviruses as primordial progenitors or ancient missing links is a bit forced and muddles the exciting discoveries. The term “missing link” in particular could lead to confusion given its use in human evolution when referring to ancestral lineages.

So overall, the logic of the writing could be cleaned up and presented more consistently and succinctly, in my opinion, and the statements about how mirusviruses clarify early evolution of NCLDV and herpesviruses should be toned down. This study is still quite impactful and describes a remarkable discovery regardless of the specific evolutionary scenario that is most likely.

-In terms of style, the writing of the manuscript could be made more focused and concise by shortening (or moving the supplementary) some analyses that do not strongly add to the manuscript. I admire that so many different analyses have been focused on the mirusviruses in an attempt to characterize this group, but some of the various Tara Oceans resources are not particularly helpful. For example, the transcriptome analysis is not critical and is perhaps a bit off-topic. It is fairly obvious that some of these genes would be expressed, and a full figure of this is not needed and merely distracts and detracts from the major point of interest (a novel chimeric lineage). Similarly, the co-occurrence analysis does not yield high confidence host predictions, so perhaps slight de-emphasis on this could help focus the manuscript.

-Throughout the manuscript the support values in the trees are usually not given (or at least I could not make out how to see them) - it would be useful if they could be integrated into the visualizations somehow so one could more easily gauge how confident the branching patterns are. Fig 3 in particular looks like it has many long branches, and confidence values would be welcome. Also, some additional rationale for the rooting of the trees in Fig 3 would be useful.

-The methods are detailed and robust. Some small questions, mainly because I am curious:
539 - which IQ-TREE models were tested? This can be done with the TEST or MFP commands, with the latter providing a wider range of models. I am curious what final models were chosen - is this information anywhere?

569 - How were the genomes clustered? Average linkage?

635 - what was the logic for using C30 as opposed to, for example, C60?

Why was a 90% ID cutoff used for transcript mapping? It seems like this is the same procedure for metagenome mapping, for which a 95% cutoff was used.

Other

Type in Figure S8 legend- “Syntheny” should be “Synteny”? How was this figure made?

Referee #3 (Remarks to the Author):

In this manuscript, the authors described a new virus lineage dubbed Mirusviricota that was

identified in ocean metagenomes of the Tara oceans project. Using extensive phylogenetic analyses and functional inference, the authors show that, despite encoding the virion module of Duplodnaviria, Mirusviricota genomes exhibit several features that are typical of Nucleocytoviricota, such as the informational module and an expanded gene repertoire that are comparable in size and function to giant viruses. The analyses are thorough and the data is readily available and organized. Overall, this is a high-quality manuscript that includes new data resources, and well supported hypotheses for the emergence of Mirusviricota that have important implications for our understanding of virus evolution.

I have a few questions and suggestions, mostly regarding the phylogenetic analysis.

- Mirusviricota is not monophyletic in the RNAPolB (Figure S1, S2), RNAPolA, and TFII (Figure S2) phylogenies. The monophyly of Mirusviricota is strongly supported by the fact that the concatenated and the MCP phylogenies are concordant, but I suggest that the authors evaluate whether the topology of these trees would change if the alignments were restricted to catalytic/conserved domains. Importantly, support values should be shown in the tree so that readers could check if the split of Mirusviricota into several clades could be a consequence of poorly-supported branches.
- Branch support should also be shown in the DNAPolB tree (Figure S4).
- Does the data in the DNAPolB tree support the models in Figure 5? Would it be possible to infer the direction of the transfer of the informational module from it? The fact that Mirusviricota/Herpesviridae is closer to the Delta clade than to Pandoravirales is contradictory to the hypothesis of a direct transfer of the informational module?
- The data resource is very organized and comprehensive, and the usage of the AGNOSTOS framework is appreciated. However, I couldn't find important pieces of data: (1) the RNAPolB HMM used for mining the metagenomes (lines 529–531); (2) the cluster memberships (although the cluster data is available in Tables S5 and S6, a reader can't reconstruct the clusters from the FASTA file without the membership data).
- Lines 174–176: Would it be possible to infer directionality by measuring the distances of the structures? Maybe a cellular homolog could be included as an outgroup.
- Lines 231–234: The alignments used to predict the structures included only the orthologs described in the supplementary tables or you conducted a database search to build the MSA? In case database searches were conducted, were the orthologs found in cellular organisms or just in metagenomes?
- Line 242: Mirusviricota and Herpesvirales also share DNAPolB, not just the virion module.
- Line 322: It is not clear what the word “versus” means here.
- Line 336: “ecological niches” is imprecise, since this sentence doesn't discuss ecology. “Broad geographic” would work better.
- Lines 366–368: Did the authors check for misassembly (e.g. using read mapping)? Does this gene contain known domains (from Pfam, ECOD, CATH, etc.)? Can you find homologs in big environmental databases such as BFD or ColabFold DB?
- Lines 700–707: Are the predicted structures reliable across most of their length? Did the authors set a minimum pLDDT to accept the structures?

Referee #4 (Remarks to the Author):

General comments:

It isn't every day that someone discovers a whole new phyla! The manuscript reports on the

discovery of a curious group of viruses discovered within the assemblies of Tara Oceans metagenome sequence libraries from size fractions > 0.2 um in size. Termed mirusviruses, these metagenome assembled genomes (MAGs) display 'missing' link characteristics between phyla within two newly described realms of DNA viruses the Duplodnaviria and the Variadnaviria. Over the past few years, the International Committee for the Taxonomy of Viruses has been busy reorganizing the taxonomic classification of viruses based on genomic characteristics. This new taxonomy is much improved and is a far better reflection of the polyphyletic nature of viral evolution. The discovery of mirusviruses strongly supports the data and evolutionary hypotheses behind ICTV classification. With this MS, along with the steady stream of recent papers surrounding ICTV taxonomy, many aspects of viral evolutionary history are finally beginning to make sense. This new taxonomy rooted in phylogenetics and structural biology will be enormously helpful in discerning genome to phenome to ecology links within unknown viruses. It is notable that all of the genomic data supporting the Mirusviricota phyla comes from metagenomes. Only a few years ago, supporting such a discovery based on environmental DNA sequencing alone would have been unthinkable. Fortunately, times have changed and the virology and microbiology communities are now comfortable with metagenome data leading the way towards discovery of completely new viral and microbial taxa. Now the exciting challenge is out to find and cultivate the first mirusvirus. Overall, I found the MS to be highly compelling, building from the strongest evidence first, with each evidence layer solidly supporting the authors proposed ideas.

The only thing I found mildly frustrating in reviewing the MS was the fact that the supplemental table filenames nor the files themselves had references to the table numbers. Thus, it was challenging connecting the supplemental tables to the text.

Specific comments:

- Text Note, page 2

Ln 78-81: So is the new class within the Nucleotyoviricota the same thing as the Mirusviricota? As written this reporting is a bit confusing.

- Text Note, page 3

In 111: I Fig. S1 it would help the reader if the authors highlighted the branch colors of the Mirus clades as was done for Fig. 1. Relying on the outer circle alone it is difficult to discern the Mirus clades of RpoB.

- Text Note, page 4

Fig. S2. lacks a legend for the rings and it is difficult to ascertain the specific hallmark gene corresponding to each ring. I would suggest adding a legend that defines each of these rings as well as the meanings of the colors in the source ring.

- Text Note, page 5

Ok, so according to PolB the Mirusviruses are closer to Herpesviruses than to Caudoviruses and others. Hence, they are a missing link.

- Text Note, page 6

Ln 217: It is slightly amazing that occurring in 50% of MAGs constitutes a “core gene cluster”! Is there an accepted heuristic for such a seemingly low frequency for a “core gene”? Were these 111 MAGs considered full length genomes, or is the low percentage simply because these are partial genomes and the actual frequency of these genes within Miruviruses is actually much higher?

- Text Note, page 7

Ln 241: Since the authors are making an argument that miruviruses show a “functional” similarity to algal viruses based on environmental observations, it would be good to see environmental metadata added to fig S5.

- Text Note, page 8

Ln 289-293: Really unclear how the co-occurrence was conducted. Such analyses can be rife with correlation issues that aren’t necessarily indicative of virus-host associations. The authors need a brief description of how these associations were obtained and they need to point out that most mirus clades showed no host association.

- Text Note, page 9

Fig. 4: Not really sure why the ribbon diagram of the jelly-roll fold protein is shown. It isn’t mentioned in the caption or the text.

- Text Note, page 10

Fig. S8: It isn’t exactly clear what the colors of the gene names mean. I am assuming that the beige colored names are informational whereas the green are viron structure, however, this information should be in the caption or legend.

- Text Note, page 11

Fig. 5: It seems like this figure could be condensed as much of panels B and C are repeated. It isn’t clear why the authors use the term “non-eukaryotic” in this figure. I think it would be more meaningful to use the 2021 ICTV taxonomy terms here for phylum (Nucleocytoviricota); class (Caudoviricetes); and order (Herpesvirales)? This is all the more true as they are using the phylum nomenclature for Miruviricota.

- Text Note, page 11

Ln 384-385: Other than terminase, Fig. S8 doesn’t seem to support the statement that miruvirus virion genes belong to the Duplodnaviria.

- Text Note, page 11

Ln 391-392: This statement assuming chimerism in the informational genes of nucleocytoviricota with a duplodnaviria origin seems like a leap. It is not at all clear how the authors come to this conclusion.

- Text Note, page 12

Ln 427-429: This claim of miruvirus abundance is only weakly supported by the data. No other data indicating the abundance of miruviruses relative to other viral groups was shown in the MS. Maybe

some else has already reported virus abundance data from analysis of Tara sequence data. The authors could refer to this analysis a reference for this statement about mirusvirus abundance.

- Text Note, page 12

Ln 439-441: The authors should cite Fig. 3 for this statement about Micromonas obtaining its heliorhodopsin from miruviruses.

- Text Note, page 13

In 482-486: The authors should cite Fig. 1 for this sentence about capsid protein structure.

- Text Note, page 21

Fig. S1 Caption: For these sorts of radial tree diagrams I believe it is better to describe the metadata displays as “rings” rather than “layers”. So it would be “inner ring, middle ring, outer ring”.

- Text Note, page 22

Fig. S2: Same comment about rings versus layers.

- Text Note, page 23

Fig. S3: What is the degree of the rotation?

- Text Note, page 24

Fig. S4: I like how the branches are colored in this tree. This approach of branch coloring should be done on the other trees.

Author Rebuttals to Initial Comments:

Reviewer number 2:

Gaia et al present a metagenomic study of large DNA viruses in the ocean. The authors use a binning approach to recover genomes of large DNA viruses, including NCLDVs, and discover a novel lineage that appears to be a chimera of NCLDV and herpesviruses. The morphogenetic module of these Mirusviruses (capsid, packaging ATPase) bears homology to Herpesviruses and tailed phages, while the rest of the genome (including the information module) is more similar to NCLDVs. Detailed bioinformatic and predicted structural analyses are performed to explore this novel lineage.

We are thankful for the time the reviewer has invested into our manuscript. Their constructive remarks in order to best position the chimeric attributes of mirusviruses in the evolutionary context of other eukaryotic viruses were particularly useful.

Overall this is an excellent study that uses creative bioinformatic methods to describe a remarkable new viral lineage that has so far eluded detection despite its widespread occurrence in the ocean. The discovery of a novel viral lineage with chimeric genomes is an important milestone, and the evolutionary analysis of mirusviruses and structural comparisons between capsids are by far the most significant components of this study. To my knowledge this is the first example of a truly chimeric dsDNA viral lineage that is the product of two distinct deep-branching lineages, and I agree with the authors that this represents a new phylum. This discovery is notable and will be of interest to a broad range of scientists, and as such warrants publication in a premier journal.

We thank the reviewer for summarizing the significance of mirusviruses, and for emphasizing the strengths of our study.

That said, I do have some general comments and constructive disagreement regarding the evolutionary interpretation that I hope the authors will consider. Below I have laid out my understanding of the facts and a possible alternative explanation for the cryptic evolutionary origins of the Mirusviruses.

We largely agree with the constructive comments made by the reviewer. They helped us to substantially improve the clarity of our manuscript. In addition to addressing each comment in the subsequent sections, here we provide a summary of our responses:

(1) The current ICTV taxonomy framework places the major capsid protein (and other virion module components) as the principal marker to define the high-level taxonomy of eukaryotic DNA viruses. This is described in detail in the first paragraph of the introduction. In our manuscript, the interpretation of the data is based on this framework. Nevertheless, we have improved clarity of the discussion by addressing comments from the reviewer. For instance, we now state that “a *Nucleocytoviricota* virus may have swapped its virion module with that of an uncharacterized duplodnavirus that co-infected the same host, while retaining the elaborate informational module.” Following this line of thought, we also extensively describe shared genes and functions between mirusviruses and giant viruses. They emphasize the unique chimeric attributes of the putative phylum ‘*Mirusviricota*’, affiliated to *Duplodnaviria* based on its virion module.

(2) We now better explain why our data points to a direct evolutionary link between herpesviruses and mirusviruses. Most critically, the two triplex proteins in the virion of these two clades are absent in *Caudoviricetes*. Predicted 3D structures for these triplex proteins have been analyzed in more details (see Figures 1 and S4, and Table S5) to support this conclusion. In addition, mirusviruses and herpesviruses are sister clades

based on the DNAPolB tree (see Figure S5). Both the results and discussion sections have also been implemented with additional text to better convey these points:

Results (Segment #1): --- *“Nevertheless, **multiple components of this module provided critical insights clarifying the evolutionary trajectory of mirusviruses. First, the two triplex capsid proteins, which form a heterotrimeric complex and stabilize the capsid shell through interactions with adjacent MCP subunits (ref 25), are conserved across herpesviruses but are missing in Caudoviricetes. Second, in herpesvirus MCPs, the HK97-fold domain, referred to as the ‘floor’ domain and responsible for capsid shell formation, is embellished with a ‘tower’ domain that projects away from the surface of the assembled capsid (ref26). The ‘tower’ domain is an insertion within the A-subdomain of the core HK97-fold (refs 26,27). In mirusviruses, the MCP protein also contains an insertion within the A-subdomain, albeit of substantially smaller size (Figures 1, S3 and S4). Such ‘tower’ domain has not been thus far described for any member of the Caudoviricetes, including the so-called jumbo phages (i.e., phages with very large genome (ref 28)). Overall, the triplex capsid proteins and the MCP ‘tower’ represent hallmark traits pointing to a closer evolutionary relationship between mirusviruses and herpesviruses compared to their bacterial and archaeal relatives.**”* ---

Discussion (Segment #2): --- *“The identification of ‘Mirusviricota’ expands the presence of duplodnaviruses beyond animals to eukaryotic plankton hosts, strongly suggesting their ancient association with eukaryotes. The presence and location of the tower domain combined with the conservation of the two triplex capsid proteins (none of these are present in known Caudoviricetes) in both ‘Mirusviricota’ and Herpesvirales (see Figure 1) strongly suggests a common ancestry of these eukaryotic viruses, rather than independent evolution from distinct Caudoviricetes clades. The deep-branching positioning of mirusvirus informational genes attesting to one or multiple ancient transfers (Figures 1 and S2) and close similarity of the DNAPolB between the two eukaryotic Duplodnaviria clades compared to other DNA virus clades (Figure S5) provide complementary information. With the shorter size of the tower domain and considering the later emergence of animals compared to unicellular eukaryotes, ‘Mirusviricota’ viruses might more closely resemble the ancestral state of eukaryotic duplodnaviruses. Thus, mirusviruses point to a planktonic ancestry for herpesviruses, which would have undergone reductive evolution, most notably, losing the transcription machinery, and specialized to the infection of animal cells (ref 37).”* ---

(3) We removed any mention of a missing link and better clarified how mirusviruses might have, under one of the two outlined hypotheses (discussion), contributed to the emergence of giant eukaryotic viruses. Overall, we are now much more balanced between the “giant virus origin” (now introduced first) and “mirusvirus origin” hypotheses for the informational module present in two distantly related viral realms. This change in wording better reflects our data and the literature. These critical changes in the discussion address critical points provided by the reviewer:

Discussion (Segment #3): --- *“Similarly enigmatic is the evolutionary trench between large and giant Nucleocytoviricota genomes and relatively simple varidnaviruses with modest gene repertoires for virion formation and genome replication (those infecting Bacteria and Archaea, as well as virophages, Adenoviridae, or else yaraviruses and polintoviruses (refs 38,39)). It has been speculated that some of these simple varidnaviruses might represent evolutionary intermediates between bacterial phages and eukaryotic viruses including the*

Nucleocytoviricota (ref 5). The genomic complexity of mirusviruses within plankton, and their core functions shared with Nucleocytoviricota provide additional insights. The informational module, and possibly other functions, may have been transferred from Nucleocytoviricota to the ancestor of mirusviruses ('giant virus origin' hypothesis), contributing to the complexification of eukaryotic duplodnaviruses. Under this scenario, a Nucleocytoviricota virus may have swapped its virion module with that of an uncharacterized duplodnavirus that co-infected the same host, while retaining the elaborate informational module. Yet, our data do not exclude the equally thought-provoking possibility of a transfer of the informational module from a mirusvirus to more simple ancestors of Nucleocytoviricota ('mirusvirus origin' hypothesis). This scenario could help explain the evolutionary leap from 'small' varidnaviruses to the overwhelmingly complex Nucleocytoviricota. Regardless of the hypothesis under consideration, mirusviruses clarify the evolutionary trajectory of eukaryotic double-stranded DNA viruses from both realms." ---

Discussion (Segment #4): --- *"While the mirusviruses likely predated the emergence of herpesviruses, the timeline for 'Mirusviricota' origins within plankton (before or after that of giant eukaryotic viruses) has yet to be elucidated." ---*

-The most dramatic claim made in this study is that Mirusviruses are an ancient progenitor to either Herpesviruses or NCLDV, depending on the evolutionary scenario (Fig 5). The evidence that mirusviruses are a primordial progenitor to either NCLDV or herpesviruses is speculative, however, and other scenarios appear more likely to me. It is not more parsimonious to presume that mirusviruses are simply NCLDV that swapped out the usual structural components for those of a different viral realm?

We have clarified the text to avoid any confusion. While the mirusviruses might more closely resemble the common ancestor of herpesviruses, they are described as sister clades (see Segments #1 and #2) throughout the manuscript.

In addition, Segment #3 now states that *"a Nucleocytoviricota virus may have swapped its virion module with that of an uncharacterized duplodnavirus that co-infected the same host, while retaining the elaborate informational module"*, thus incorporating the essence of the reviewer's comment. As mentioned above, we have followed the ICTV taxonomy framework in our manuscript. As a result, even under this scenario the mirusviruses would remain affiliated to the realm *Duplodnaviria*.

There are several reasons why this more recent emergence of mirusviruses appears more likely (to me at least): 1) Other NCLDVs have also acquired different capsids (pithoviruses, pandoraviruses) so this is actually rather common- the only difference here is that they acquired a HK97 capsid and associated genes.

Our data indicates that mirusviruses are an ancient and distinct lineage: they are clearly separated from other *Nucleocytoviricota* classes based on the phylogenomic analysis of the informational module (Figure 1, strong support values) and possess many core genes entirely missing in *Nucleocytoviricota* (Figure 3). In addition, the phylogenies of their concatenated informational and virion markers are globally congruent, despite the relatively large number (>100) of genomes (Figure 2). This strongly suggests that the core informational module transfer(s) occurred long ago, and that mirusviruses maintain a unique functional lifestyle compared to *Nucleocytoviricota*. The origin of these core genes, or else the nearly 10,000 mirusvirus singleton genes, is unknown. As a result, even though we observed more shared functions between mirusviruses and

Nucleocyotiviricota compared to herpesviruses (this is extensively described in the main text), most of the genomic makeup of mirusviruses could not be linked to any other known virus lineage.

In addition, the timing of the informational module transfer is posterior to the emergence of mirusviruses, since this clade is defined based on the virion module (MCP fold) and not the information module (see comments above). Thus, in our view the emergence of mirusviruses cannot be limited to the acquisition of giant virus genes. That being said, we have no relevant data to test which of the two clades is the oldest, and as a result this point is not a focus of our study and remains unclear (see Segment #4). We are hoping that more genomic recoveries for ‘*Mirusviricota*’ in the oceans and beyond will help clarify this question.

Finally, none of the known giant virus lineages have replaced the pre-existing structural module with an unrelated one and at least some of the components of the ancestral structural module are retained. In poxviruses, the major capsid protein was repurposed for the function of scaffolding protein, whereas the I7-like maturation protease and the genome packaging ATPase are still functioning in the same capacity; in pandoraviruses, the one of the major structural proteins has evolved from preexisting minor structural protein (PMID: 33686356), whereas the packaging ATPase is still conserved; in pithoviruses, conversely, the ATPase was lost, but the DJR MCP is still encoded. Thus, the case of mirusviruses, which lack any of the *Nucleocyotiviricota* virion module components is very different.

2) Phylogenies of genes of the information module show that mirusviruses usually branch from within NCLDV (Fig S2), indicating that NCLDV already existed and had diversified by the time that Mirusviruses emerged. The PolB tree with eukaryotic homologs (Fig S4) provides some tentative evidence of an independent Mirus origin, but this is not definitive because NCLDV have acquired this gene from eukaryotes multiple times themselves.

We have improved the individual-gene phylogenies (see Figures S2 and S5) and provided key support values in those trees as well as in the concatenated tree (Figure 1 and Figure S2). We slightly modified the text to clarify these results:

--- ***“Single-gene phylogenies place these MAGs in one (DNAPolB) or multiple clades (RNAPolA and RNAPolB), always in between the known Nucleocyotiviricota orders (Figure S2). Signal for TFIIS was weaker due to its shorter length. Robust phylogenomic inferences of the concatenated four informational gene markers indicate that they represent a monophyletic viral clade with several hallmark genes closely related to, yet distinct from those in the known Nucleocyotiviricota classes (Figure 1).”*** ---

On the one hand, the fact that mirusviruses are not monophyletic based on RNAPolA and RNAPolB trees can be used to favor a “giant virus origin” for the informational module with transfers occurring after the emergence of the known *Nucleocyotiviricota* classes. On the other hand, the more robust phylogenomic analysis using the entire information module (4 genes) strongly supports the monophyly of mirusviruses. This is coherent with the functional clustering analysis placing the mirusviruses together (Figure S7). Nevertheless, the single-gene phylogenies are mentioned in the manuscript, albeit only displayed in supplemental figures.

We agree with the reviewer that the DNAPolB phylogeny (now substantially improved) alone cannot be used to identify the origin of the informational module. As discussed above, individual trees provide limited information. In our study, we only used the

DNApolB phylogeny to support the phylum-level novelty of mirusviruses and provide some context regarding their link to herpesviruses. But again, due to the absence of an objective outgroup, the positioning of mirusviruses outside of any known class of giant viruses prevents us from knowing the origin of the informational module between the realms *Duplodnaviria* and *Varidnaviria*.

3) The phylogenetic breadth of the mirusviruses is not as broad as NCLDV, suggesting that they represent an order or class level of NCLDV that swapped out the morphogenetic module at some point.

Clearly, the known *Nucleocytoviricota* viruses are substantially more diverse compared to the known mirusviruses. This point is emphasized in the main text and in the Figure 1. However, members of the *Nucleocytoviricota* were discovered decades ago and have been extensively studied in many biomes and by many research groups. In contrast, mirusviruses have thus far only been explored in the surface of the oceans using a DNA-dependent RNA polymerase as a guide for binning. It is possible that many more mirusvirus lineages (in other ecosystems, or within the oceans but lacking the DNA-dependent RNA polymerase gene) will be discovered in the years to come. Only after a more thorough global search will it be relevant, in our view, to compare the overall breadth of diversity for *Nucleocytoviricota* and mirusviruses. Regardless, such comparison is beyond the scope of our current survey, which used an original methodology to identify previously overlooked marine RNApolB clades and explore their genomic context, ultimately leading to the discovery of chimeric viruses forming a putative new phylum.

4) The other similarities of Mirusviruses and NCLDV, such as the common presence of histones, rhodopsins, and other functional genes also points to the overall NCLDV origin of mirusviruses. Indeed, the gene sharing tree also shows Mirusviruses emerging from within NCLDV

Some of the mirusvirus genes are closely related to *Nucleocytoviricota* and the ancestral status of histones and rhodopsins has been clarified (see next section). However, and as described above, most of the mirusvirus core genes, or else the nearly 10,000 mirusvirus singleton genes are not found among the *Nucleocytoviricota*. Thus, the dendrogram for gene sharing tree, which is not a phylogeny, cannot be used as a strong argument to argue that mirusviruses emerged from within the *Nucleocytoviricota*. The analysis is of interest because it emphasized the important functional similarities between the two groups, contrasting with the differences between mirusviruses and herpesviruses. Finally, the functions at the interface of virus-host interaction, such as rhodopsins, are subject to extensive and repeated horizontal exchange between viruses as well as between viruses and hosts, and as such probably reflect a shared host/environment rather than ancestral relationship. This is true for mirusviruses but also for *Nucleocytoviricota* (e.g., rhodopsins are hardly to be expected in animal poxviruses or asfarviruses).

5) The rhodopsins and histones of Mirusviruses don't appear to be particularly basal branching compared to Mirusviruses- it's a bit hard to see because the NCLDV branches are not colored, but I don't see evidence of Mirusvirus ancestry here.

Indeed, there is no particular evidence for the ancestral status of mirusviruses based on results from the two phylogenies. In addition, the evolution of these two functions does not recapitulate the evolution of *Nucleocytoviricota* orders. In order to keep a strong focus on the most relevant data to describe the importance of mirusviruses, we decided to put these phylogenetic analyses as larger supplemental figures (now Figure S8 and S9,

support values have been added). We have colored the branches and named all clades affiliated to the main mirusvirus subclades and *Nucleocytoviricota* orders to improve clarity. In addition, we have updated the text to clarify this important point:

--- ***“Phylogenetic inferences of the histones and rhodopsins point to a complex evolutionary history of these genes in both ‘Mirusviricota’ and Nucleocytoviricota, with multiple horizontal transfer events between the virus clades and marine planktonic eukaryotes (Figures S8 and S9). In addition, a Micromonas heliorhodopsin may have originated from a mirusvirus (Figure S9), suggesting that ‘Mirusviricota’ contributes, alongside Nucleocytoviricota (refs 3,4), to the evolution of planktonic eukaryotes by means of gene flow.”*** ---

Now, the scenario in which mirusviruses are a primordial progenitor to herpesviruses is a bit more plausible, though I still do not favor this hypothesis. The main evidence for this is that the mirusvirus capsid is highly divergent from herpesvirus capsids, but still contains the tower domain. But this pattern could arise through many different processes - not necessarily only because mirusviruses gave rise to herpesviruses. For example, we don't know the evolutionary rates / mutation rate of mirusviruses - it may simply be higher than herpesviruses and NCLDV, which could explain a more rapid divergence. If this was the case, mirusviruses may have emerged from within herpesviruses (coinfection of a eukaryotic host may have been the initial site for recombination). Similarly, if the floor domain evolved convergently, then mirusviruses may have evolved from tailed phages, without any real link to herpesviruses. Based on the present data it is not clear if any of these alternative scenarios can be confidently ruled out.

As discussed in previous sections, we provided more data and have clarified the evolutionary link between herpesviruses and mirusviruses, which are described as sister clades (a brief summary is presented in the Figure 4, panel A). We have now better introduced our main arguments pointing to herpesviruses and mirusviruses sharing a common eukaryotic virus ancestor (see Segments #1 and #2 above). Regarding the critical virion module, our arguments focus on the triplex proteins, with the MCP tower as a supporting shared trait. We agree that the tower alone was a speculative argument, but in combination with the two shared triplex proteins, which are not present in *Caudoviricetes*, the possibility of convergent evolution becomes highly unlikely. In addition, the two lineages are sister clades in the improved DNAPolB phylogeny (Figure S5). Besides, there is no reason to suspect a higher mutation rate for mirusviruses compared to herpesviruses or NCLDVs. Indeed, it has been established that mutation rate correlates with the genome size and is similar for viruses with dsDNA genomes (Gago et al., 2009, Science; PMID: 19265013).

The reviewer also wonders if the mirusviruses could have originated from herpesviruses. Our data does not favor this possibility: (1) the DNAPolB clearly separates mirusviruses and herpesviruses, (2) mirusviruses infect unicellular eukaryotes in the oceans while known herpesviruses only infect animals (a relatively recent eukaryotic clade), (3) and our phylogenomic and phylogenetic analyses (see replies above) strongly advocate for the mirusviruses to be very ancient.

-In my view, it would be helpful if more space and discussion were given to the significance of the Mirus capsid structure and other morphogenetic components (terminase, etc). This is undoubtedly the most important part of this paper, and the strongest evidence of genome chimerism, so a more detailed explanation of the floor vs tower domains could be given, together with their evolutionary significance. Is it possible

that the tower domain could evolve convergently due to shared function in eukaryotic viruses? Does the structure necessarily imply shared evolutionary history? What else is known about the role of the tower domain? Right now the link between Mirusviruses and herpesviruses is somewhat tentative given the inability to produce molecular phylogenies. Predicted structures of the herpesvirus, phage, and mirusvirus packaging ATPase would also be welcome in the main text, given this is a major crux of the paper.

We thank the reviewer for stressing the importance of the virion module hallmark genes, which are indeed a critical component of the study that is challenged by their considerable divergences between *Caudoviricetes*, herpesviruses and mirusviruses.

First, the Segment #1 (see above) clarifies the importance of the triplex proteins. Second, we have now produced more predicted 3D protein structures for each hallmark gene of the virion module among diverse families of mirusviruses, herpesviruses and *Caudoviricetes*. Results (blastp, Foldseek) are presented in the new Figure S4 and Table S5. Triplex capsid proteins were also incorporated into the main Figure 1.

-Perhaps I missed it, but it would be useful if the authors could define the level of divergence between the herpesvirus, phage, and mirusvirus morphogenetic components a bit more. Is there no detectable homology of the mirusvirus proteins using BLASTP, Psiblast, or HMMER3? Perhaps an alignment could be made and put in the supplementary just to show that aligned blocks do not exist? The lack of detectable sequence homology between the mirusvirus proteins and other homologs is a key point of this paper, so adding something else in addition to statements in the text would be helpful.

This information was overlooked in our initial submission. We now have performed comprehensive blast comparisons between mirusvirus, *Caudoviricetes* and herpesvirus genes, and have quantified the protein-level similarity. Results have been compiled in Table S5 (each hallmark gene), and in the Figure S4 (HK97 MCP only).

Besides, the lack of proper alignments for the virion morphogenetic module was already known when comparing herpesviruses and *Caudoviricetes* (e.g., see <https://www.nature.com/articles/s41598-019-47742-z>). Our own alignments confirm this lack of informative signal, and are now part of the data availability.

-66-71: the whole premise of the ambiguity of how NCLDV emerged from smaller viruses is somewhat overstated, in my opinion. Many related smaller viruses have Double jelly roll fold capsids - namely polintoviruses, virophages, and most recently Yaravirus (10.1073/pnas.2001637117)- and they already have been proposed to be a link between NCLDV and smaller viruses (10.1016/j.virol.2015.02.039). The Woo et al study that is cited on line 70 nicely shows this too. Even if this premise were correct, it is not clear how mirusviruses necessarily solve this riddle given that their genomes are rather large (the complete genome is 435 kbp) - they are not really an intermediate between NCLDV and smaller viruses. Once again, I feel that the logic pushing mirusviruses as primordial progenitors or ancient missing links is a bit forced and muddles the exciting discoveries. The term “missing link” in particular could lead to confusion given its use in human evolution when referring to ancestral lineages.

The intermediate position of mirusviruses is not that much about genome length, but rather their nature of duplodnaviruses, related to the *Caudoviricetes* and their highly variable genomic lengths, yet sharing the hosts, environments and many genes with the only Varidnaviria phylum that reaches similar (and larger) lengths. It was hence meant as an overall status, but we agree that this was somewhat confusing. We removed

“missing link”, which was initially only used in the abstract. We also clarified statements regarding how mirusviruses could have played a key role in the emergence of giant viruses and the manuscript is now more balanced between the “giant virus origin” (now introduced first) and ‘mirusvirus origin’ hypotheses for the informational module (see Segments #2 and #3 above). In particular, Segments #3 incorporated references and other points raised by the reviewer. The clarity of our discussion has been substantially improved.

So overall, the logic of the writing could be cleaned up and presented more consistently and succinctly, in my opinion, and the statements about how mirusviruses clarify early evolution of NCLDV and herpesviruses should be toned down. This study is still quite impactful and describes a remarkable discovery regardless of the specific evolutionary scenario that is most likely.

As mentioned above, we have now clarified the evolutionary significance of mirusviruses (see segments #1, #2 and #3 above). We thank the reviewer for emphasizing the importance of the mirusvirus discovery regardless of the specific evolutionary scenario.

-In terms of style, the writing of the manuscript could be made more focused and concise by shortening (or moving the supplementary) some analyses that do not strongly add to the manuscript. I admire that so many different analyses have been focused on the mirusviruses in an attempt to characterize this group, but some of the various Tara Oceans resources are not particularly helpful. For example, the transcriptome analysis is not critical and is perhaps a bit off-topic. It is fairly obvious that some of these genes would be expressed, and a full figure of this is not needed and merely distracts and detracts from the major point of interest (a novel chimeric lineage). Similarly, the co-occurrence analysis does not yield high confidence host predictions, so perhaps slight de-emphasis on this could help focus the manuscript.

We agree with the reviewer that it is important to keep a focus on the evolutionary prominence of mirusviruses. We removed the host-prediction analyses but maintained the metatranscriptomic analysis in the main text as it provides two critical pieces of information. First, genes for the virion module are much more expressed compared to the informational module. Since these two modules are critical components of our manuscript, this information is highly relevant. Second, the figure shows the size fractions in which mirusviruses are most active. The data links mirusviruses to unicellular eukaryotes, which is highly relevant in our argumentation about the evolutionary relationship of mirusviruses and animal-infecting herpesviruses (see third paragraph of the discussion). As a result, we have maintained the figure as a main figure, and tried to shorten the Results sections on the abundance and activity of mirusviruses as much as possible, without overlooking prime information to contextualize their ecological importance.

-Throughout the manuscript the support values in the trees are usually not given (or at least I could not make out how to see them) - it would be useful if they could be integrated into the visualizations somehow so one could more easily gauge how confident the branching patterns are. Fig 3 in particular looks like it has many long branches, and confidence values would be welcome. Also, some additional rationale for the rooting of the trees in Fig 3 would be useful.

We apologize for the lack of support values in the initial submission, which were only presented in the figure 2. We have added those values for the critical branches in figure 1 as well as in supplemental figures S2, S5, S8 and S9.

-The methods are detailed and robust. Some small questions, mainly because I am curious:

539 - which IQ-TREE models were tested? This can be done with the TEST or MFP commands, with the latter providing a wider range of models. I am curious what final models were chosen - is this information anywhere?

The models were tested through the MFP command. We apologize for the lack of information in the first version of our manuscript. The model selected for each tree is now indicated in the legend of the corresponding figures.

569 - How were the genomes clustered? Average linkage?

We applied a single linkage method. If MAG A is connected to MAG B, and if MAG B is connected to MAG C, then the MAGs A, B, and C are within the same cluster.

635 - what was the logic for using C30 as opposed to, for example, C60?

The logic of using C30 instead of C60 for the large concatenation phylogeny (1,722 sequences; 3,715 sites) was essentially based on computational burden, despite using the PMSF framework. Indeed, we considered critical to include a freerate (+R), which from our experience tends to generate more resolute trees even with less categories of mixture models. This option however substantially increases the memory required for each increment of number of categories. Therefore, the main tree was based on the LG+C30+F+R10 (PMSF). We have now obtained a nearly identical tree with the LG+C60+I+G (PMSF), albeit with slightly lower support values. As a result, we have kept results from our initial strategy for this manuscript.

Why was a 90% ID cutoff used for transcript mapping? It seems like this is the same procedure for metagenome mapping, for which a 95% cutoff was used.

A 95% cutoff was used during the binning procedure (see "Constrained automatic binning with CONCOCT"), as well as to assess the distribution (see "Biogeography of the GOEV database") and activity (see "Metatranscriptomics of the GOEV database") of genomes in the GOEV database. We thank the reviewer for identifying this typo. We have corrected this important value in the Methods section.

Other

Type in Figure S8 legend- "Syntheny" should be "Synteny"? How was this figure made?

The typo has been corrected. The figure was made using anvi'o v7.1 program "anvi-interactive" with the "manual-mode" as a flag.

Again, we thank the reviewer for the very thoughtful comments especially on the side of the evolutionary complexity of mirusviruses. Their comments have substantially improved the quality of the manuscript.

Reviewer number 3:

In this manuscript, the authors described a new virus lineage dubbed Mirusviricota that was identified in ocean metagenomes of the Tara oceans project. Using extensive phylogenetic analyses and functional inference, the authors show that, despite encoding the virion module of Duplodnaviria, Mirusviricota genomes exhibit several features that are typical of Nucleocytoviricota, such as the informational module and an expanded gene repertoire that are comparable in size and function to giant viruses. The analyses are

thorough and the data is readily available and organized. Overall, this is a high-quality manuscript that includes new data resources, and well supported hypotheses for the emergence of Mirusviricota that have important implications for our understanding of virus evolution.

We thank the reviewer for emphasizing some of the strengths of our manuscript, and for providing highly relevant comments that helped us further improve our manuscript.

I have a few questions and suggestions, mostly regarding the phylogenetic analysis.

- Mirusviricota is not monophyletic in the RNAPolB (Figure S1, S2), RNAPolA, and TFII (Figure S2) phylogenies. The monophyly of Mirusviricota is strongly supported by the fact that the concatenated and the MCP phylogenies are concordant, but I suggest that the authors evaluate whether the topology of these trees would change if the alignments were restricted to catalytic/conserved domains. Importantly, support values should be shown in the tree so that readers could check if the split of Mirusviricota into several clades could be a consequence of poorly-supported branches.

Those are indeed very relevant points.

First, the figure S1 (now extensively improved to add in clarity) includes all RNAPolB from the TARA assemblies, which include fragments, duplicates, and long branching artifacts that were not solved at that early stage, because the point of this figure was solely to identify deep-branching clades lacking representatives (the “RNAPolB new clades”), without having to resolve their deep-branching evolutionary history.

As the reviewer noted, the phylogenomic analysis in Figure 1 (main support values have been added) is more relevant to clarify the evolutionary trajectory of mirusviruses as compared to trees for individual genes. Importantly, this phylogenomic analysis (alike trees in Figure S2) was performed on a curated dataset from which the duplicated sequences for each hallmark gene and the long-branch artifacts have been removed.

Regarding the Figure S2, we have now improved each single-gene phylogeny and added support values for the key branches. Different approaches have been tried following the reviewer’s comment: notably including structural homologs to the alignments, trimming the sites based on the conservation of their entropy (with the BMGE software; in practice this reduced the alignments to their most conserved segments), trimming uninformative sites (with ClipKIT), or trimming sites with different thresholds of gaps. Most produced trees with similarly low supports. Eventually, nothing proved as positively impactful as iteratively resampling the taxa to remove long branches and unstable/ambiguous sequences, which is the strategy that was used for the new single-protein trees presented in Fig S2. Even if the supports remain relatively low at deep nodes, they are higher than before and the trees more stable.

For the single-protein tree of the DNAPolB with viral and eukaryotic homologs (Figure S5), the approach that proved to be the most efficient in improving the tree was to carefully remove long and/or unstable branches, and the least phylogenetically informative sequences with Treemmer based on an average tree length of 0.95.

- Branch support should also be shown in the DNAPolB tree (Figure S4).

We apologize for this oversight and have added the support values for all key branches in the DNAPolB tree, as well as in other figures and supplemental figures.

- Does the data in the DNAPolB tree support the models in Figure 5? Would it be possible to infer the direction of the transfer of the informational module from it? The fact that Mirusviricota/Herpesviridae is closer to the Delta clade than to Pandoravilares is contradictory to the hypothesis of a direct transfer of the informational module?

The evolutionary history of the DNAPolB was already known to be more complex as compared to RNAPolA and RNAPolB. In our analysis for instance, the DNAPolB of Pokkesviricetes is distant from those of other Nucleocytoviricota genomes. Thus, our view is that unfortunately, individual trees such as those in figures S2 (RNAPolA, RNAPolB, DNAPolB and TFIIS) and S5 (extended DNAPolB analysis including more viruses and the eukaryotes, which has now been improved and is more robust) cannot solve the direction of transfer for the informational module.

As suggested in previous articles, the informational module likely coevolved between protoeukaryotes and their eukaryotic viruses. The tree topology is in line with ancient transfers involving the common ancestor of mirusviruses and herpesviruses, proto-eukaryotes, and giant eukaryotic viruses. But again, the evolutionary history of the DNAPolB is considered complex, and we see it as hazardous to claim a direction of transfer based on the DNAPolB tree topology alone. We now have a better-balanced wording to describe the two main hypotheses, in order to address comments made by this reviewer and others:

--- *“Similarly enigmatic is the evolutionary trench between large and giant Nucleocytoviricota genomes and relatively simple varidnaviruses with modest gene repertoires for virion formation and genome replication (those infecting Bacteria and Archaea, as well as virophages, Adenoviridae, or else yaraviruses and polintoviruses (refs 38,39)). It has been speculated that some of these simple varidnaviruses might represent evolutionary intermediates between bacterial phages and eukaryotic viruses including the Nucleocytoviricota (ref 5). The genomic complexity of mirusviruses within plankton, and their core functions shared with Nucleocytoviricota provide additional insights. The informational module, and possibly other functions, may have been transferred from Nucleocytoviricota to the ancestor of mirusviruses (‘giant virus origin’ hypothesis), contributing to the complexification of eukaryotic duplodnaviruses. Under this scenario, a Nucleocytoviricota virus may have swapped its virion module with that of an uncharacterized duplodnavirus that co-infected the same host, while retaining the elaborate informational module. Yet, our data do not exclude the equally thought-provoking possibility of a transfer of the informational module from a mirusvirus to more simple ancestors of Nucleocytoviricota (‘mirusvirus origin’ hypothesis). This scenario could help explain the evolutionary leap from ‘small’ varidnaviruses to the overwhelmingly complex Nucleocytoviricota. Regardless of the hypothesis under consideration, mirusviruses clarify the evolutionary trajectory of eukaryotic double-stranded DNA viruses from both realms.” ---*

- The data resource is very organized and comprehensive, and the usage of the AGNOSTOS framework is appreciated. However, I couldn't find important pieces of data: (1) the RNAPolB HMM used for mining the metagenomes (lines 529–531); (2) the cluster memberships (although the cluster data is available in Tables S5 and S6, a reader can't reconstruct the clusters from the FASTA file without the membership data).

Regarding the RNAPolB HMM, our figshare link in the initial submission (data availability section) provided this information, however the naming of the file (HMMs_NCLDVs) lacked clarity, for which we apologize. We now have improved this by renaming the file

“HMMs” and re-organizing its content as follow: “00_HMM_RNApolB” (contains the RNApolB HMM that allowed discovery of mirusviruses), and “01_HMM_Nucleocytoviricota_hallmark_genes” (contains HMMs for various hallmark genes). As mentioned in the text, those HMMs were designed in a previous study led by some of the authors (<https://www.pnas.org/doi/10.1073/pnas.1912006116>).

Regarding the lack of AGNOSTOS information, the table S6 (sheet #1) provides access to the occurrence of 29,413 gene clusters across all genomes in the GOEV database. However, critically missing was the corresponding gene IDs in the genomes. This has now been included as a separate supplemental table: Table S7. That table links the unique IDs of genes, contigs genomes and gene clusters in a comprehensive table. The associated data is also available from the figshare link in “Data availability”. We thank the reviewer for helping improve the relevance of supplemental tables and the data availability.

- Lines 174–176: Would it be possible to infer directionality by measuring the distances of the structures? Maybe a cellular homolog could be included as an outgroup.

This point relates to the MCP 3D structure. We have now predicted more 3D structures for the MCP of mirusvirus, herpesvirus and *Caudoviricetes* families (Table S5 and Figure S4). We have applied FoldSeek to those predicted 3D structures in order to compare this structure set. Results perfectly recapitulated the three clades (see dendrogram in Figure S4, panel A). Given the interesting results this analysis provided, we have expanded this approach to other hallmark genes of the *Duplodnaviria* virion module. Results are summarized in Table S5. However, it is not possible to root the structure-based trees, because there are no cellular homologs for the *Duplodnaviria* structural proteins, complicating the inferences regarding the directionality of transfer.

We thank the reviewer for suggesting this highly relevant analysis, which allowed to perform protein comparisons despite the lack of proper alignments for phylogenies.

- Lines 231–234: The alignments used to predict the structures included only the orthologs described in the supplementary tables or you conducted a database search to build the MSA? In case database searches were conducted, were the orthologs found in cellular organisms or just in metagenomes?

Alignments were done only using the 0.6 million genes in GOEV, based on the AGNSOTOS framework. For the 10 core genes mentioned in these lines, they occurred only in mirusviruses within the GOEV, except for one core gene that occurs to a lesser extent among the giant viruses (as seen in Figure 3). These ten core genes lacked similarities to reference databases at the structural level using DALI SEARCH (see our method sections), but also at the protein sequence level using NCLB blast. Thus, some of the mirusvirus core gene clusters appear to have no homologous genes in known cellular organisms.

- Line 242: Mirusviricota and Herpesvirales also share DNApolB, not just the virion module.

This sentence lacked clarity and we have removed mention of the virion module. The text now reads as follow:

--- *“Thus, function-wise mirusviruses more closely resemble the Nucleocytoviricota viruses (many of which are also widespread at the surface of the oceans, see Figure 1) as compared to Herpesvirales.”* ---

- Line 322: It is not clear what the word “versus” means here.

This was an error and we have now deleted “(e.g., surface samples versus)” from the legend. We thank the reviewer for identifying this typo.

- Line 336: “ecological niches” is imprecise, since this sentence doesn’t discuss ecology. “Broad geographic” would work better.

We agree and have removed the mention of “ecology”. The text now reads as follow:

--- “Mirusviruses have different **biogeographic distributions**” ---

- Lines 366–368: Did the authors check for misassembly (e.g. using read mapping)? Does this gene contain known domains (from Pfam, ECOD, CATH, etc.)? Can you find homologs in big environmental databases such as BFD or ColabFold DB?

We have performed extra analyses for this gene and found weak signal for bacterial genes related to pathogenicity (YadA bacterial adhesin protein / collagen-like adhesin known to have repetitive structures). However, this gene is not part of the core mirusvirus genes and its length prevented us from predicting its 3D structure thus far. In order to keep a focus on the broad evolution of mirusviruses, and in light of the reviewer comments, we decided to remove mention of this gene in the main text.

- Lines 700–707: Are the predicted structures reliable across most of their length? Did the authors set a minimum pLDDT to accept the structures?

We did not set a minimum pLDDT to accept the structures, however the AF2 model was good across most of the length:

Plot: IDDT amino acid positions for the mirusvirus MCP presented in Figure S3.

Once again, we would like to thank the reviewer for the constructive comments and for expressing relevant concerns that altogether substantially improved the quality of our manuscript.

Reviewer number 4:

General comments:

It isn’t every day that someone discovers a whole new phyla! The manuscript reports on the discovery of a curious group of viruses discovered within the assemblies of Tara Oceans metagenome sequence libraries from size fractions > 0.2 um in size. Termed mirusviruses, these metagenome assembled genomes (MAGs) display ‘missing’ link characteristics between phyla within two newly described realms of DNA viruses the Duplodnaviria and the Variadnaviria. Over the past few years, the International

Committee for the Taxonomy of Viruses has been busy reorganizing the taxonomic classification of viruses based on genomic characteristics. This new taxonomy is much improved and is a far better reflection of the polyphyletic nature of viral evolution. The discovery of mirusviruses strongly supports the data and evolutionary hypotheses behind ICTV classification. With this MS, along with the steady stream of recent papers surrounding ICTV taxonomy, many aspects of viral evolutionary history are finally beginning to make sense. This new taxonomy rooted in phylogenetics and structural biology will be enormously helpful in discerning genome to phenome to ecology links within unknown viruses. It is notable that all of the genomic data supporting the Mirusviricota phyla comes from metagenomes. Only a few years ago, supporting such a discovery based on environmental DNA sequencing alone would have been unthinkable. Fortunately, times have changed and the virology and microbiology communities are now comfortable with metagenome data leading the way towards discovery of completely new viral and microbial taxa. Now the exciting challenge is out to find and cultivate the first mirusvirus. Overall, I found the MS to be highly compelling, building from the strongest evidence first, with each evidence layer solidly supporting the authors proposed ideas.

We thank the reviewer for their positive comments, and for placing our discovery in the broad context of ICTV taxonomy and the quest to understand viral evolution.

The only thing I found mildly frustrating in reviewing the MS was the fact that the supplemental table filenames nor the files themselves had references to the table numbers. Thus, it was challenging connecting the supplemental tables to the text.

We apologize for this situation. It is likely the submission process failed to correctly link those tables, which are in an Excel format incompatible with PDF transformation, and thus have been submitted using a possibly inadequate approach. All our supplemental tables start with their number. For example, Table S1 is named "Table_S1_939_metagenomes". We hope our second submission will be more compatible with these types of supplemental tables. Alternatively, the figshare link in our data availability section provides access to the latest version of all supplemental tables.

Specific comments:

- Text Note, page 2

Ln 78-81: So is the new class within the Nucleotytoviricota the same thing as the Mirusviricota? As written this reporting is a bit confusing.

To avoid any confusion and to keep the focus on the main discovery, we removed mention for the new *Nucleocyotviricota* class. The segment now reads as follow (change in bold):

--- "*We characterized and manually curated hundreds of population genomes **that expand the known diversity of Nucleocyotviricota**. But most notably, our survey led to the discovery of plankton-infecting relatives of herpesviruses that form a putative new phylum we dubbed 'Mirusviricota'.*" ---

- Text Note, page 3

In 111: I Fig. S1 it would help the reader if the authors highlighted the branch colors of the Mirus clades as was done for Fig. 1. Relying on the outer circle alone it is difficult to discern the Mirus clades of RpoB.

We have now substantially improved the readability of Figure S1:

Figure S1: Identification of novel DNA-dependent RNA polymerase B (RNAPolB) clades in the sunlit ocean. The maximum-likelihood phylogenetic tree is based on 2,728 RNAPolB sequences more than 800 amino acids in length with similarity <math><90\%</math> (gray color in the **inner ring**) identified from 11 large marine metagenomic co-assemblies. This analysis also includes 262 reference RNAPolB sequences (red color in the **inner ring**) corresponding to known archaeal, bacterial, eukaryotic and giant virus lineages for perspective. The **middle ring** shows the number of RNAPolB sequences from the 11 metagenomic co-assemblies that match to the selected amino acid sequence with identity >90% (log10). The **outer ring displays selections made for the different clades**. Finally, RNAPolB new lineages are labelled with a red dot for mirusviruses (families were characterized in subsequent analyses) and in blue for Proculviricetes.

We colored the branches and emphasized the RNAPolB new clades with large dots. We also collapsed a large clade encompassing all RNAPolB genes with a bacterial origin, which we used for rooting the tree. This collapse allows for a better view of especially the Mirus clades, for which we linked the corresponding family ID as determined in our subsequent investigations (e.g., see Figure 2). We thank the reviewer for the suggestion, which substantially improved clarity of this important first supplemental figure.

- Text Note, page 4

Fig. S2. lacks a legend for the rings and it is difficult to ascertain the specific hallmark gene corresponding to each ring. I would suggest adding a legend that defines each of these rings as well as the meanings of the colors in the source ring.

We have simplified the figures to keep a focus on the phylogenetic topologies (which have been improved by removing long branching artefacts). Especially, rings for the *Nucleocyotoviricota* hallmark genes were removed since they overlap with the Figure 1.

For clarity, we also colored the branches as a function of the taxonomy. Finally, we added support values for the main branches.

- Text Note, page 5

Ok, so according to PolB the Mirusviruses are closer to Herpesviruses than to Caudoviruses and others. Hence, they are a missing link.

Indeed, the DNAPolB of mirusviruses is closer to the main clade of herpesviruses, compared to *Caudovirales*.

- Text Note, page 6

Ln 217: It is slightly amazing that occurring in 50% of MAGs constitutes a “core gene cluster”! Is there an accepted heuristic for such a seemingly low frequency for a “core gene”? Were these 111 MAGs considered full length genomes, or is the low percentage simply because these are partial genomes and the actual frequency of these genes within Mirusviruses is actually much higher?

Many of the mirusvirus MAGs are likely incomplete (average length of 200 kb compared to the >400 kb near-complete contiguous genome introduced at the end of the results sections). As a result, we used a flexible cut-off for the core gene clusters. We are confident that more stringent cut-offs will be used after we have access to additional near-complete genomes. Despite the current limitations, using a 50% cut-off was highly relevant in our investigations. Especially, it allowed us to identify by means of projected 3D structures multiple components of the virion morphogenetic module among the mirusviruses. This included the MCP, but also the two triplex proteins that were only known to be present in herpesviruses prior to the discovery of mirusviruses.

- Text Note, page 7

Ln 241: Since the authors are making an argument that mirusviruses show a “functional” similarity to algal viruses based on environmental observations, it would be good to see environmental metadata added to fig S5.

The segment was confusing. The functional similarity is based on gene clusters, not on environmental observations. It shows a strong link between mirusviruses and reference *Nucleocytoviricota* genomes from culture. Since a majority of known environmental *Nucleocytoviricota* genomes occur in the surface of the oceans (see Figure 1, or Table S10), we wanted to make a point that the two clades display functional similarities and share the same ecosystem. To improve clarity and to address this comment, we have removed mention of plankton in the Figure S6, and modified the text as follow:

--- “Clustering of ‘Mirusviricota’ MAGs and reference viral genomes from culture (including *Nucleocytoviricota*, *Herpesvirales* and *Caudoviricetes*) based on quantitative occurrence of gene clusters highlighted the strong functional differentiation between mirusviruses and herpesviruses and, conversely, a strong functional similarity between mirusviruses and *Nucleocytoviricota* (Figure S6 and Table S9). Thus, function-wise mirusviruses more closely resemble the *Nucleocytoviricota* viruses (many of which are also widespread at the surface of the oceans, see Figure 1) as compared to *Herpesvirales*.” ---

We thank the reviewer for improving the clarity of this segment.

- Text Note, page 8

Ln 289-293: Really unclear how the co-occurrence was conducted. Such analyses can be rife with correlation issues that aren’t necessarily indicative of virus-host associations.

The authors need a brief description of how these associations were obtained and they need to point out that most mirus clades showed no host association.

We agree and have entirely removed the host-prediction analysis, which only provided signal for a small subset of mirusvirus clades, as emphasized by the reviewer. We now have a stronger focus on the evolutionary prominence of mirusviruses, as requested by another reviewer.

- Text Note, page 9

Fig. 4: Not really sure why the ribbon diagram of the jelly-roll fold protein is shown. It isn't mentioned in the caption or the text.

We have removed the predicted 3D structure from the figure (now Figure 3).

- Text Note, page 10

Fig. S8: It isn't exactly clear what the colors of the gene names mean. I am assuming that the beige colored names are informational whereas the green are virion structure, however, this information should be in the caption or legend.

This is correct, and we have added this important information in the supplemental figure (now Figure S11).

- Text Note, page 11

Fig. 5: It seems like this figure could be condensed as much of panels B and C are repeated. It isn't clear why the authors use the term "non-eukaryotic" in this figure. I think it would be more meaningful to use the 2021 ICTV taxonomy terms here for phylum (Nucleocytoviricota); class (Caudoviricetes); and order (Herpesvirales)? This is all the more true as they are using the phylum nomenclature for Mirusviricota.

We improved Panel A and have also simplified the Panels B and C to minimize repetition (now Figure 4). We also used the ICTV taxonomy, as suggested.

- Text Note, page 11

Ln 384-385: Other than terminase, Fig. S8 doesn't seem to support the statement that mirusvirus virion genes belong to the Duplodinaviria.

The near-complete mirusvirus genome contains all hallmark genes identified thus far for the virion morphogenesis module of *Duplodnaviria*: a HK97-fold MCP, the two triplex proteins, the portal protein, the terminase, and capsid maturation protease. Thus, we think this figure (now Figure S11) does support the statement. The caption that has been added to the figure (see previous comment) should clarify this point.

- Text Note, page 11

Ln 391-392: This statement assuming chimerism in the informational genes of nucleocytoviricota with a duplodnaviria origin seems like a leap. It is not at all clear how the authors come to this conclusion.

This sentence lacked clarity and was deleted.

- Text Note, page 12

Ln 427-429: This claim of mirusvirus abundance is only weakly supported by the data. No other data indicating the abundance of mirusviruses relative to other viral groups was shown in the MS. Maybe some else has already reported virus abundance data from

analysis of Tara sequence data. The authors could refer to this analysis a reference for this statement about mirusvirus abundance.

The biogeographic signal for mirusviruses and *Nucleocytoviricota* in the GOEV database are described in detail in a dedicated table (now Table S10), and their occurrence across the TARA Oceans size fractions are part of Figure 1. However, the analysis comparing the relative abundance of mirusviruses and *Nucleocytoviricota* clades was lacking. To fill this gap, we have now summarized the cumulative mean coverage of each GOEV genome across the TARA Oceans metagenomes (genomes with no signal were excluded to avoid bias from cultured *Nucleocytoviricota* genomes characterized from other ecosystems). We integrated this analysis into Table S10 and generated a new supplemental figure (Figure S10):

Figure S10: Environmental signal of virus eukaryotic clades in the sunlit ocean. For each marine eukaryotic virus clades, the box plots display cumulative mean coverage of GOEV genomes among 937 TARA Oceans metagenomes. Only genome detected in at least one metagenome were considered. The number of considered genomes per clade and their cumulative coverage median are also described.

We have incorporated these metrics into the main text, as follows (changes in **bold**):

--- "To our knowledge, 'Mirusviricota' represents the first eukaryote-infecting lineage of Duplodnaviria found to be widespread and abundant within plankton in the sunlit oceans. Indeed, mirusviruses were detected in 131 out of the 143 TARA Oceans stations, from pole to pole. They occurred mostly in the 0.2-5 μm (76.3% of the entire mirusvirus metagenomic signal) and 3-20 μm (15.4%) size fractions that cover a high diversity of unicellular planktonic eukaryotes²² (Figures 1 and 2, Table S10). **Among the TARA Oceans metagenomes considered in our study, the total mean coverage of marine Nucleocytoviricota MAGs and culture genomes in GOEV was 15 times higher compared to the mirusvirus MAGs, reflecting the current imbalance in genomic units between these two phyla (1,706 vs. 111). Yet, median cumulative mean coverage for the mirusviruses was higher compared to viruses in all Nucleocytoviricota orders, with the noticeable exception of Algavirales (Figure S10 and Table S10). Thus, the mirusviruses are among the most abundant eukaryotic viruses currently characterized in the sunlit oceans.**

The mirusviruses are not only abundant, but also highly active within plankton. In fact, the mirusvirus MAGs, which contain just 3.8% of genes in GOEV, represent 13% of the TARA Oceans metatranscriptomic signal for this genomic database (Table S11). This substantial in situ transcriptomic signal stresses the relevance of 'Mirusviricota' to eukaryotic virus-host dynamics in marine systems." ---

We thank the reviewer for helping us emphasizing the environmental importance of mirusviruses, in the context of already well characterized *Nucleocytoviricota* orders.

Ln 439-441: The authors should cite Fig. 3 for this statement about *Micromonas* obtaining its heliorhodopsin from miruviruses.

For clarity, this statement was moved to the Results section to focus solely on key points in the discussion:

--- *"In addition, a *Micromonas heliorhodopsin* may have originated from a miruvirus (Figure S9), suggesting that 'Mirusviricota' contributes, alongside Nucleocytoviricota (refs 3,4), to the evolution of planktonic eukaryotes by means of gene flow."* ---

- Text Note, page 13

Ln 482-486: The authors should cite Fig. 1 for this sentence about capsid protein structure.

We now cite Figure 1 to support the statement on MCP.

- Text Note, page 21

Fig. S1 Caption: For these sorts of radial tree diagrams I believe it is better to describe the metadata displays as "rings" rather than "layers". So it would be "inner ring, middle ring, outer ring".

We modified the legend accordingly.

- Text Note, page 22

Fig. S2: Same comment about rings versus layers.

We modified the legend accordingly.

- Text Note, page 23

Fig. S3: What is the degree of the rotation?

The structures were rotated around the x axis by $\sim 90^\circ$.

- Text Note, page 24

Fig. S4: I like how the branches are colored in this tree. This approach of branch coloring should be done on the other trees.

Indeed, this adds in clarity. While colors were already parts of Figures 1 and 2, we have now added branch coloring in the figures for the RNAPolB (Figure S1), histones, heliorhodopsins, as well as for the individual trees in Figure S2. Note that for the referred figure (previously Figure S4, now Figure S5), we opted for a collapsing of most clades to add in clarity.

We thank the reviewer for their constructive comments, and for improving the clarity and overall quality of our manuscript.

Reviewer Reports on the First Revision:

Referee #2 (Remarks to the Author):

The authors have addressed my concerns, and I believe the logic of the different evolutionary scenarios is easier to follow now. The addition of the information on the triplex capsid proteins is welcome and provides useful context. This is an impressive effort and a milestone in our understanding of viral diversity, especially in the context of how new viral lineages can emerge through chimerism. Personally I still favor the "giant virus hypothesis" as the most parsimonious, but I understand the need to present alternatives as well.

A small note on wording- it sometimes sounds awkward to say things like "Caudoviricetes and Nucleocytoviricota viruses" (as in on line 73) - perhaps in these cases it is easier to reword to "viruses within the Caudoviricetes and Nucleocytoviricota".

Referee #3 (Remarks to the Author):

The revised version of the manuscript is much improved and addressed all of my major points. The significant expansion of the structural comparisons is welcomed and provide strong evidence for the hypothesis presented in the text. I do not have any major suggestions.

Minor points:

- Please include the PDB files for the generated structures in the figshare. These structures support the main hypothesis and should be available for users. If possible, also provide the pIDDT plots.
- I thank the authors for renaming the RNAPolB HMM file. It is much clearer what that file represents now. I suggest that, if possible, the authors upload the original MSAs used to generate the HMMs. HMMER-formatted HMMs limit their usage for a single tool (HMMER) and hide information from the original MSA.

Referee #4 (Remarks to the Author):

The authors have done a through job of responding to each reviewer's comments. I have no further comments.

Author Rebuttals to First Revision:

Reviewer number 2:

The authors have addressed my concerns, and I believe the logic of the different evolutionary scenarios is easier to follow now. The addition of the information on the triplex capsid proteins is welcome and provides useful context. This is an impressive effort and a milestone in our understanding of viral diversity, especially in the context of how new viral lineages can emerge through chimerism. Personally I still favor the "giant virus hypothesis" as the most parsimonious, but I understand the need to present alternatives as well.

A small note on wording- it sometimes sounds awkward to say things like "Caudoviricetes and Nucleocytoviricota viruses" (as in on line 73) - perhaps in these cases it is easier to reword to "viruses within the Caudoviricetes and Nucleocytoviricota".

We modified the sentence accordingly, and we thank the reviewer for their substantial contributions.

Reviewer number 3:

The revised version of the manuscript is much improved and addressed all of my major points. The significant expansion of the structural comparisons is welcomed and provide strong evidence for the hypothesis presented in the text. I do not have any major suggestions.

Minor points:

- Please include the PDB files for the generated structures in the figshare. These structures support the main hypothesis and should be available for users. If possible, also provide the pIDDT plots.

The mentioned PDB files are available from the Figshare link, and this is stated in the Data availability statement. In addition, the overall pIDDT scores for each PDB file are summarized in the Table S5 (see sheet "Metadata & 3D structure quality"). Finally, we have included the pIDDT plots for each PDB file in the Figshare, as requested.

- I thank the authors for renaming the RNAPolB HMM file. It is much clearer what that file represents now. I suggest that, if possible, the authors upload the original MSAs used to

generate the HMMs. HMMER-formatted HMMs limit their usage for a single tool (HMMER) and hide information from the original MSA.

We have now added the alignment file, and we thank the reviewer for their contributions to our study.

Reviewer number 4:

The authors have done a through job of responding to each reviewer's comments. I have no further comments.

We thank the reviewer for contributing to the strength of our study.